



# OH level populations and accuracies of Einstein-A coefficients from hundreds of measured lines

Stefan Noll[1,2], Holger Winkler[3], Oleg Goussev[2], and Bastian Proxauf[4]

[1]Institut für Physik, Universität Augsburg, Augsburg, Germany
[2]Deutsches Fernerkundungsdatenzentrum, Deutsches Zentrum für Luft- und Raumfahrt, Weßling-Oberpfaffenhofen, Germany
[3]Institut für Umweltphysik, Universität Bremen, Bremen, Germany
[4]Max-Planck-Institut für Sonnensystemforschung, Göttingen, Germany

**Correspondence:** S. Noll (stefan.noll@dlr.de)

**Abstract.** OH airglow is an important nocturnal emission of the Earth's mesopause region. As it is chemiluminescent radiation in a thin medium, the population distribution over the various roto-vibrational OH energy levels of the electronic ground state is not in local thermodynamic equilibrium (LTE). In order to better understand these non-LTE effects, we studied hundreds of OH lines in a high-quality mean spectrum based on observations with the high-resolution Ultraviolet and Visual Echelle

Spectrograph at Cerro Paranal in Chile. Our derived populations cover vibrational levels between $v = 3$ and 9, rotational levels up to $N = 24$, and individual $\Lambda$-doublet components when resolved. As the reliability of these results critically depends on the Einstein-A coefficients used, we tested six different sets and found clear systematic errors in all of them, especially for Q-branch lines and individual $\Lambda$-doublet components. In order to minimise the deviations in the populations for the same upper level, we used the most promising coefficients from Brooke et al. (2016, JQSRT 168, 142) and further improved them with an empirical

correction approach. The resulting rotational level populations show a clear bimodality for each $v$, which is characterised by a probably fully thermalised cold component and a hot population where the rotational temperature increases between $v = 9$ and 4 from about 700 to about 7,000 K and the corresponding contribution to the total population at the lowest $N$ decreases by an order of magnitude. The presence of the hot populations causes non-LTE contributions to rotational temperatures at low $N$, which can be estimated quite robustly based on the two-temperature model. The bimodality is also clearly indicated

by the dependence of the populations on changes in the effective emission height of the OH emission layer. The degree of thermalisation decreases with increasing layer height due to a higher fraction of the hot component. Our high-quality population data are promising with respect to a better understanding of the OH thermalisation process.

## 1 Introduction

The nighttime emission of the Earth's atmosphere in the near-infrared is dominated by hydroxyl (OH) airglow (Meinel, 1950;

Rousselot et al., 2000; Hanuschik, 2003; Noll et al., 2012, 2015), which originates in the mesopause region in a layer with a width of about 8 km and a typical peak height of 87 km (Baker and Stair, 1988). The various bright roto-vibrational bands of the OH electronic ground state $X^2\Pi$ represent an important tracer for atmospheric dynamics (especially wave propagation), ambient temperatures, and chemical composition (especially atomic oxygen) at these high altitudes, which are mostly probed





by ground- and satellite-based remote sensing (e.g. Taylor et al., 1997; Beig et al., 2003; von Savigny et al., 2012; Mlynczak
et al., 2013; Reisin et al., 2014; Sedlak et al., 2016; Noll et al., 2017). For these applications, it is crucial to understand the
physical mechanisms that lead to the observed line emission.

In the mesopause region, OH is mostly formed by the reaction of hydrogen and ozone (Bates and Nicolet, 1950; Xu et al.,
2012), which excites the electronic ground state up to the ninth vibrational level $v$ (Charters et al., 1971; Llewellyn and Long,
1978; Adler-Golden, 1997). The nascent population distribution over the roto-vibrational levels is far from local thermody-
namic equilibrium (LTE). As the subsequent relaxation processes by collisions with other atmospheric species are relatively
slow compared to the radiative lifetimes of the excited states (e.g., Adler-Golden, 1997; Xu et al., 2012; Kalogerakis et al.,
2018; Noll et al., 2018b), the OH emission bands (which contribute to the vibrational relaxation) reveal strong non-LTE effects.
The vibrational level populations can be fitted as a function of energy by an exponentially decreasing (i.e. Boltzmann-like) dis-
tribution with a pseudo-temperature of around 10,000 K (Khomich et al., 2008; Noll et al., 2015; Hart, 2019a). Hence, OH
bands with upper state vibrational levels $v'$ up to the highest nascent state can easily be measured. Moreover, the rotational
level populations for the different $v$ reveal high overpopulations for high rotational states $N$ compared to the lowest three or
four levels under the assumption of a thermal distribution (Pendleton et al., 1989, 1993; Dodd et al., 1994; Cosby and Slanger,
2007; Oliva et al., 2015; Noll et al., 2018b). The pseudo-temperatures for the high-$N$ populations achieve values up to those
found for the $v$ levels (Oliva et al., 2015). The theoretical explanation of these populations especially for low $v$ is still uncertain
(Dodd et al., 1994; Kalogerakis et al., 2018; Noll et al., 2018b) as their modelling suffers from limitations in the data sets and
uncertain input parameters (especially rate coefficients for collisional transitions). It is usually assumed that the ratios of lines
related to the lowest $N$ of a fixed $v$ are sufficiently close to LTE for a reliable estimate of the ambient temperature (e.g. Beig
et al., 2003). However, this assumption appears to be insufficient at least for the highest $v$, where deviations of several kelvins
were found (Noll et al., 2016, 2018b). In addition, small modifications in the set of considered levels in terms of $N$ can already
significantly change the corresponding population temperature (Noll et al., 2015).

A successful study of OH level populations requires accurate molecular parameters, i.e. line wavelengths, level energies, and
Einstein-A coefficients. In particular, the latter suffer from relatively high uncertainties despite numerous dedicated studies for
their calculation (e.g., Mies, 1974; Langhoff et al., 1986; Turnbull and Lowe, 1989; Nelson et al., 1990; Goldman et al., 1998;
van der Loo and Groenenboom, 2007; Brooke et al., 2016) and evaluation (e.g., French et al., 2000; Pendleton and Taylor,
2002; Cosby and Slanger, 2007; Liu et al., 2015; Hart, 2019b). Apart from the derivation of absolute OH level populations or
densities (Noll et al., 2018b; Hart, 2019b), the quality of these transition probabilities especially affects OH-based temperature
estimates (Liu et al., 2015; Noll et al., 2015; Parihar et al., 2017; Hart, 2019b) and abundance retrievals for species like atomic
oxygen (Mlynczak et al., 2013; Noll et al., 2018b). The persistent uncertainties in the Einstein-A coefficients are obviously
related to the molecular structure of OH and the lack of adequate data for the calculation of the molecular parameters (Nelson
et al., 1990; Pendleton and Taylor, 2002; Cosby and Slanger, 2007; van der Loo and Groenenboom, 2007; Brooke et al., 2016).

In order to improve our knowledge on OH level populations and Einstein-A coefficients, high-quality measurements of
a large number of OH lines and a detailed analysis are required. We could perform such a study based on high-resolution
spectroscopic data taken with the Ultraviolet and Visual Echelle Spectrograph (UVES; Dekker et al., 2000) at the Very Large





Telescope at Cerro Paranal in Chile (24.6° S, 70.4° W). A mean spectrum of the highest quality spectra (totalling 536 hours

of exposure time) allowed us to investigate 723 lines with upper vibrational levels $v'$ between 3 and 9 in the optical and near-infrared regime in detail. In many cases, the small $\Lambda$ doubling effect due to rotational-electronic perturbations between the ground and excited electronic states (Pendleton and Taylor, 2002) was resolved.

In Sect. 2, we describe the UVES data set. Then, we discuss the data analysis involving the calculation of the mean spectrum, the measurement of line intensities, and a check of the line positions (Sect. 3). Section 4 discusses the differences in the derived

OH level populations for the Einstein-A coefficients of Mies (1974), Langhoff et al. (1986), Turnbull and Lowe (1989), van der Loo and Groenenboom (2008), Rothman et al. (2013), and Brooke et al. (2016). Moreover, the latter are used as the basis for an empirical improvement of the coefficients. The corresponding OH level populations are then investigated in detail (Sect. 5). This involves population fitting, a study of the non-LTE contributions to rotational temperatures, and the investigation of population differences caused by a change in the OH emission altitude. Finally, we draw our conclusions (Sect. 6).

## 2   Data set

This study is based on so-called Phase 3 products of the astronomical echelle spectrograph UVES (Dekker et al., 2000) provided by the European Southern Observatory. Noll et al. (2017) selected about 10,400 archived spectra taken between April 2000 and March 2015, extracted the night-sky emission, and performed a complex flux calibration procedure in order to investigate long-term variations in the mesopause region based on OH emission. The studied spectra comprise the wavelength range between

570 and 1040 nm covered by two set-ups centred on 760 and 860 nm. Depending on the width of the entrance slit, the spectral resolving power varied between 20,000 and 110,000. Hence, these data are well suited for OH level population studies as they allow one to measure numerous resolved emission lines.

As the exposure time (between 1 and 125 min) and the contamination of the night-sky emission by the astronomical target (the slit length is only between 8 and 12″) are also strongly varying, it is important to focus on spectra of sufficient quality,

especially if very weak lines are studied. The final sample of Noll et al. (2017), who studied relatively bright P-branch lines related to low rotational levels, included 3,113 suitable spectra. We use an even smaller subsample of 2,299 spectra as the basis for this study. It is related to the investigation of the faint $K(D_1)$ potassium line at 769.9 nm with a mean intensity of about 1 R (rayleigh) by Noll et al. (2019). For that sample, the selected spectra were carefully checked around the $K(D_1)$ line. In order to be able to measure even fainter lines in the entire wavelength regime and to have a homogeneous data set for the

calculation of a mean spectrum, we further reduced the sample. 45 spectra of the set-up centred on 860 nm were rejected as they showed severe flaws (wrong continuum levels) below 730 nm. This was not a problem for the potassium study. Moreover, we increased the minimum exposure time from 10 to 45 min and reduced the maximum continuum limit around $K(D_1)$ from 100 to 40 R nm$^{-1}$. Finally, we only considered spectra that were taken with the standard slit width of 1″, which corresponds to a resolving power of 42,000. 63% of the sample of Noll et al. (2019) was taken with this slit width.

The resulting sample consists of 533 high-quality spectra with a total exposure time of 536 hours at a telescope with a diameter of the primary mirror of 8 m.



## 3  Analysis

### 3.1  Mean spectrum

In order to calculate the probably best high-resolution airglow mean spectrum in the covered wavelength regime so far, we first

mapped the 533 selected spectra (Sect. 2) with the flux calibration of Noll et al. (2017) applied to a common wavelength grid from 560 to 1061 nm with a step size of 1 pm that well samples airglow lines, which have a full width at half maximum of about 20 pm close to 800 nm. The mapping is necessary since each UVES spectrum has its own wavelength grid. The set-up positioning appears to have an uncertainty of the order of 1 nm. Moreover, the original step sizes vary from 1.8 to 5.2 pm depending on the central wavelength of the set-up (760 or 860 nm), the chip (two chips with spectra separated by a small gap

at the central wavelength), and the pixel binning. Pixel pairs in dispersion direction on the chips were merged for 85% of the sample. The rest of the data is unbinned. Before the mean calculation, the spectra were also scaled to be representative of the zenith by using the van Rhijn correction (van Rhijn, 1921) for a thin layer at an altitude of 90 km. As the zenith angles at the mid-exposure times vary from 3 to 64°, this is a crucial correction with factors between 0.46 and 1.00. These factors do not significantly change across the entire OH emission layer, i.e. the choice of the reference altitude is not critical.

The mean spectrum was calculated by means of a pixel-dependent $\sigma$-clipping approach in order to avoid the contribution of strong sporadic outliers due to technical issues or the contamination by an astronomical target. As the threshold was set to 10 standard deviations, statistical noise and natural variations of the airglow emission do not cause the rejection of a spectrum at a certain pixel. The final number of considered spectra at each wavelength after three iterations of the $\sigma$-clipping is displayed in the lower panel of the upper plot in Fig. 1. The clipping only reduced the numbers by a few spectra. There is a trend towards

more rejections at longer wavelengths. The plot also reveals the impact of the combination of the two set-ups centred on 760 and 860 nm with 231 and 302 spectra, respectively. The gap between the spectra of the two chips of each set-up and very narrow gaps between the spectral orders at long wavelengths can also be seen. The smoothing of the sample-related steps in the histogram reflects the variation in the wavelength positioning of a certain set-up.

The mean spectrum in the upper plot in Fig. 1 shows 15 OH bands marked by the upper and lower vibrational levels $v'$ and

$v''$. Bands with $\Delta v = v' - v''$ between 3 and 6 are covered. The band strength strongly increases from OH(5-0) to OH(4-1) as bands with higher $v'$ and lower $\Delta v$ tend to be stronger in the covered wavelength regime. Each band is split into the three branches R, Q, and P, which are characterised by changes of the rotational quantum number $N$ of $-1$, 0, and $+1$. While the R and Q branches at the short-wavelength side and central part of the band are relatively compact, the P branch shows relatively wide spaces between the lines. P-branch lines with high upper rotational quantum numbers $N'$ are located at distinctly longer

wavelengths than those with low $N'$, i.e. they are found in regions that are dominated by other OH bands. For this reason, high-$N'$ P-branch lines of the faint OH(7-1) band can also be detected in the UVES mean spectrum.

The lower plot of Fig. 1 shows the narrow wavelength range between 727 and 745 nm to demonstrate the good spectral resolution. The plotted range includes the full Q branch and the P branch up to lines with $N' = 5$ of OH(8-3), a band of intermediate strength. The plot clearly shows the splitting of each rotational state by spin–orbit coupling. The $Q_1$ and $P_1$ lines

($F = 1$) related to the electronic substate $X^2\Pi_{3/2}$ are well separated from the fainter $Q_2$ and $P_2$ lines ($F = 2$) of $X^2\Pi_{1/2}$. For



**Figure 1.** UVES mean spectrum in rayleighs per picometre. The full spectral range with labelled OH bands is shown at the top. At the bottom, a narrower wavelength range focussing on the Q and P branch of OH(8-3) is plotted. The lines are labelled. The wavelength-dependent number of spectra involved in the calculation of the mean spectrum is indicated in additional subpanels. This number varies due to the use of two set-ups centred on 760 and 860 nm (with some variation) and the $\sigma$-clipping procedure for the mean calculation.





the visible lines, the value of $F$ does not change during the transition, i.e. $F'' = F'$. The intercombination lines which show a change of $F$ are much fainter and are therefore neglected in this study. The spectral resolving power of 42,000 is sufficiently high for seeing $\Lambda$ doubling. The separation of both components can already be found for $Q_1$ and $P_1$ lines with relatively low $N'$. In the lower plot of Fig. 1, the largest separation is visible for $Q_1(N' = 4)$. It amounts to 55 pm (Brooke et al., 2016),

i.e. this $\Lambda$ doublet is fully resolved. Separations of more than 200 pm are measurable for $P_1$ lines with $N' \geq 11$. The faintest marked $\Lambda$ doublets $Q_2(2)$ and $Q_1(4)$ have intensities between 1 and 2 R. They can easily be measured in the UVES mean spectrum, which allows one to also detect lines that are more than one order of magnitude fainter (Sect. 3.2). For a general overview of lines (not only OH) that can be accessed with UVES data, see the catalogue of Cosby et al. (2006). It is based on the night-sky atlas of Hanuschik (2003), which involves UVES observations with a total exposure time of 9 hours in the red

and near-infrared wavelength range.

### 3.2    Line intensities

As it is the most comprehensive list of calculated OH lines so far, we used the line wavelengths of Brooke et al. (2016) for the identification of lines in the UVES mean spectrum (Sect. 3.1) and the derivation of their intensities. For this purpose, the calculated vacuum wavelengths were converted into air wavelengths by means of the formula of Edlén (1966) for standard air,

which works well for the UVES data. The default line integration range was set to a width of about 2 resolution elements of the spectrograph (Sect. 2), which is 40 pm at 860 nm, plus the separation of the two $\Lambda$-doublet components. If the latter was wider than the 2 resolution elements, the components were measured independently. For an optimal continuum subtraction, the two continuum points for a linear interpolation across the line were defined manually as this approach can better handle contaminations by nearby emissions and absorptions of other lines than an automatic procedure, which was also tested. The

wavelengths of the selected continuum points were also used as limits of the integration range. In particular, range modifications were necessary for lines at very long wavelengths around 1 $\mu$m, where the spectrograph causes extended line wings. The resulting line intensities representing the zenith (Sect. 3.1) were also corrected for molecular absorption in the lower atmosphere. The complex procedure involving high-resolution radiative transfer calculations and water vapour measurements in the astronomical target spectra is described by Noll et al. (2017). Further details are given by Noll et al. (2015). For the correction

of the measured intensities, the derived line transmission values for the 533 individual spectra were averaged.

As illustrated in Fig. 1, the mean spectrum is composed of UVES spectra of two different set-ups centred on 760 and 860 nm. The wavelength shift between both set-ups causes changes in the data properties depending on wavelength. In order to minimise the impact of these changes on the measured line intensities, we investigated and corrected two effects: long-term variations in the OH line intensity and flux calibration errors. The former are important since the two UVES set-ups cover very

different parts of the sample-related period from May 2000 to July 2014. Before December 2004, there were only observations with the 860 nm set-up (Noll et al., 2017). On the other hand, spectra of this set-up are only present in the selected sample until May 2010. This results in mean 10.7 cm solar radio fluxes (Tapping, 2013) for an averaging period of 27 days of 102 solar flux units (sfu) for the 760 nm set-up and 140 sfu for the 860 nm set-up. According to Noll et al. (2017) (also based on UVES data), the mean solar cycle effect for $v'$ between 5 and 9 is $16.1 \pm 1.9\,\%$ per 100 sfu. There is no significant change





with $v'$. We took this mean percentage, the set-up-specific mean solar radio fluxes, and the wavelength-dependent fraction of 760 and 860 nm spectra to correct the OH line intensities to be representative of the mean solar radio flux of the full sample of 123 sfu. The intensity corrections were up to a few per cent with line-dependent differences characterised by a standard deviation of 2.2%. We did not consider the impact of a possible linear long-term trend as it is not significant for the UVES data (Noll et al., 2017). In order to test the flux calibration, we calculated mean spectra for each set-up and also measured line

intensities. The latter was performed automatically by using the same wavelengths for the line integration as in the case of the mean spectrum of the full sample. The continuum was measured in narrow intervals (about 0.25 resolution elements, i.e. 5 pm wide at 860 nm) around these limiting positions. The wavelength-dependent intensity ratios for the two set-ups were then used to derive correction factors depending on set-up and chip. Taking the wavelength range between the set-up gaps around 760 and 860 nm as the reference, we found correction factors close to 1 with a standard deviation of 2.7%, which is consistent with

the relative flux calibration uncertainty of about 2% for the UVES data set reported by Noll et al. (2017). Combining the solar activity and flux calibration correction, the resulting standard deviation is only 1.7% as both effects partly cancel out.

The quality of the final line intensities was indicated by a flag consisting of a primary and a secondary classifier. Each classifier is represented by a digit between 0 and 3. A value of 3 corresponds to a reliable measurement of the entire $\Lambda$ doublet, which requires symmetric line emission and a featureless underlying continuum, 1 and 2 refer to reliable measurements only for

the $\Lambda$-doublet component with e or f parity in the upper state, and 0 marks uncertainties for both components. For unresolved doublets, only 0 and 3 are possible digits. The introduction of the secondary classifier allows for a finer classification scheme. For example, the combined classes 30 and 03 can be used for ambiguous cases. Reasons for measurement uncertainties are obvious or possible blends with other emission lines, regions of significant absorptions in the continuum (often combined with very low transmission at the position of the line), and insufficient signal-to-noise ratio in the case of very weak lines.

Figure 2 shows a histogram of the measured lines depending on the decadal logarithm of the intensity in rayleighs. In total, 723 $\Lambda$ doublets are included. This neglects 13 measurements with digit 0 for the primary and secondary classifier. These lines were not further used in this study. Other potential lines could not be measured due to a blend with a stronger line, no detection, or a line wavelength within the order gaps in the near-infrared (Fig. 1). The measured intensities of the 723 $\Lambda$ doublets range from 0.01 to 600 R, i.e. they comprise almost 5 orders of magnitude. Intensities around 1 R are most abundant.

The median is 1.7 R. There is a conspicuous drop in the occurrence frequency below about 0.2 R, which suggests a strongly increasing incompleteness of detections for fainter lines. The intensity of weaker $\Lambda$ doublets also tends to be more uncertain as the intensity distributions for the different quality classes show. For the classes 3, 1+2, and 0, the median intensities are 2.6, 0.83, and 0.086 R. 546 doublets or 76% belong to class 3, where the two components were measured independently in 34% of the cases. There are 122 doublets (17%) with only one reliable component (1+2). Finally, there are 55 cases (8%) with class

0, 75% of them with resolved doublets. Considering that detached components require independent line measurements (350 cases), the total number of measurements for the data in Figure 2 amounts to 1,073.

The 723 studied $\Lambda$ doublets probe 236 different upper states characterised by $v'$, $N'$, and $F'$. Up to nine doublets contribute to the population data for a certain level. The distribution of level energies $E'$ depending on $v'$ is shown in Fig. 3. The energies range from 10,211 to 28,051 $cm^{-1}$. Except for $v' = 3$, where $N'$ only up to 9 could be measured (mainly due to the wavelength





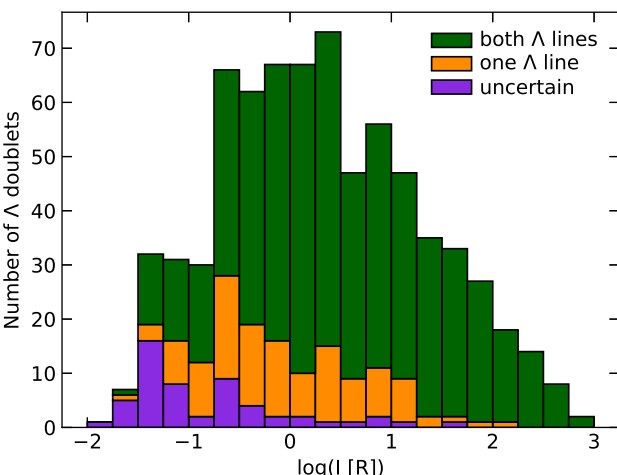

**Figure 2.** Distribution of the decadal logarithm of the measured OH Λ-doublet intensities in rayleighs. Three categories are indicated: reliable measurement of both Λ-doublet components (detached or unresolved) in green (class 3), reliable measurement of only one component in orange (classes 1 and 2), and uncertain intensities in violet (class 0 without 00).

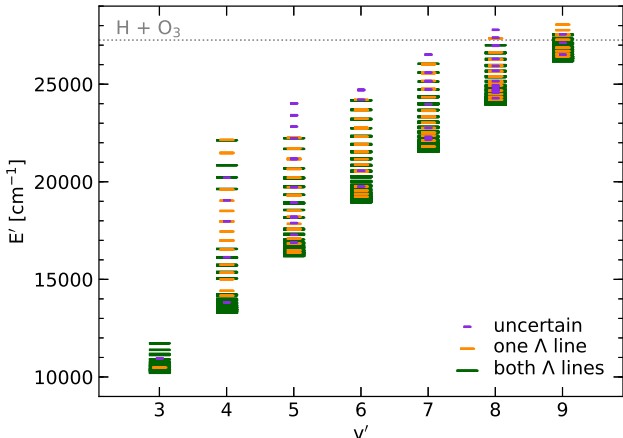

**Figure 3.** Distribution of the upper level energies $E'$ of the measured roto-vibrational transitions in inverse centimetres separated for each upper vibrational level $v'$. The same categories as in Fig. 2 are marked by different colours and line lengths (see legend). The likely exothermicity limit of the hydrogen–ozone reaction is marked by the grey dotted line.

limitations of the UVES data), wide ranges of $E'$ are covered by the data for the different $v'$. A maximum energy range of 8,861 cm$^{-1}$ is achieved for $v' = 4$. This is possible due to $N'$ up to 24. The energy ranges shrink for higher $v'$ due to a steeper decrease of the line intensities with increasing $N'$, which reduces the detectability of high-$N'$ lines. An important reason for this is certainly the closer exothermicity limit of the hydrogen-ozone reaction, which produces the excited OH. Nevertheless, there are nine levels above this limit if we assume 3.38 eV (Cosby and Slanger, 2007), i.e. about 27,260 cm$^{-1}$ (Noll et al.,





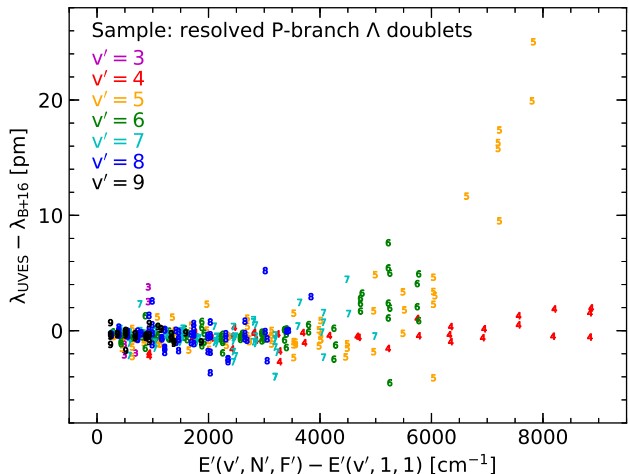

**Figure 4.** Wavelength difference between the UVES-based measurement of isolated P-branch lines (resolved Λ doublets) and the corresponding position provided by Brooke et al. (2016) (similar to HITRAN) in picometres as a function of the energy of the upper state of the transition $E'$ with respect to the lowest energy of the corresponding vibrational level $v'$ in inverse centimetres. The coloured markers indicate $v'$.

2018b). This suggests that the kinetic energy involved in the reaction is also important to populate the OH roto-vibrational levels. For the excitation of the highest level found ($v' = 9$, $N' = 12$, $F' = 1$), about $800\,\mathrm{cm^{-1}}$ of additional energy would be needed. Figure 3 also provides the primary quality classes for the lines related to the displayed states. 79% of the levels are covered by at least one line with class 3. Exclusive class 0 contribution is found for only 16 states. However, excluding uncertain lines can reduce the $E'$ range for a given $v'$. In particular, the maximum $N'$ for $v' = 5$ shows a decrease from 23 to 20 in this case.

### 3.3 Line positions

The high resolving power of 42,000 of the UVES data used allows a check of the quality of the input line positions, which were taken from Brooke et al. (2016) and were converted to standard air using the formula of Edlén (1966). The default positioning of the integration ranges for the line intensity measurements described in Sect. 3.2 worked well in most cases. However, significant shifts were necessary for some high-$N'$ lines. For a systematic study of these offsets, we took the manually adapted integration windows to calculate the intensity-weighted centroid wavelength for each line.

In Fig. 4, we show the difference between observed and model wavelengths in picometres as a function of upper state energy (neglecting the vibrational energy) for 406 individually measured Λ-doublet components of the P branch. This selection rejects unresolved or only partly resolved Λ doublets and lines with uncertain central wavelengths due to blending with other lines. The remaining 66 Q-branch and 67 R-branch Λ-doublet components are not plotted as they only probe $\Delta E'$ up to $5{,}000\,\mathrm{cm^{-1}}$ and are essentially consistent with the P-branch lines, which tend to have higher signal-to-noise ratios. The plot shows for all





$v'$ a very good agreement of observed and modelled wavelengths in the case of low energies. For $\Delta E'$ lower than $2,000\,\mathrm{cm}^{-1}$, the mean value and standard deviation are $-0.4$ and $0.7\,\mathrm{pm}$, respectively. The systematic offset is much less than the original pixel size in the UVES spectra (Sect. 3.1). Hence, it can be caused by uncertainties in the wavelength calibration. Moreover,

the assumption of standard air conditions ($1013\,\mathrm{hPa}$, $288\,\mathrm{K}$, and no $H_2O$) for the UVES instrument might cause a part of the offset. Beyond $3,000\,\mathrm{cm}^{-1}$, the displayed wavelength offsets show an increasing scatter. In part, this is caused by the higher measurement uncertainties for the fainter lines, but there are also clear trends depending on $v'$. In general, the difference between the measured and theoretical line wavelengths increases with $\Delta E'$. This increase appears to be stronger for higher $v'$. While the change for $v' = 4$ is only about $1\,\mathrm{pm}$ at around $8,000\,\mathrm{cm}^{-1}$, it is about $20\,\mathrm{pm}$ for $v' = 5$. For higher $v'$ (at least for

6 and 7), the increase of the offsets appears to be even stronger. However, as the covered energy range decreases as well, the maximum offsets only amount to a few picometres. Hence, only the measured shifts for $v' = 5$ and $N'$ of 22 and 23 are of the order of a spectral resolution element. For all other detected lines, the quality of the theoretical line positions is much better.

The discussed results are for the line wavelengths published by Brooke et al. (2016). As the HITRAN line database (Gordon et al., 2017) is more frequently used, we also calculated the wavelength shifts for these data. We took the version HITRAN2012

(Rothman et al., 2013), which does not differ from the more recent version HITRAN2016 (Gordon et al., 2017) in terms of the OH data. The results are very similar to those in Fig. 4. Strong deviations above $10\,\mathrm{pm}$ are found for the same small sample of lines, although OH(5-1)P$_1$(23) is missing in the HITRAN database. The mean difference between the line wavelengths from Rothman et al. (2013) and Brooke et al. (2016) is $0.07\,\mathrm{pm}$. The standard deviation only amounts to $0.40\,\mathrm{pm}$.

Based on the UVES mean spectrum of Hanuschik (2003) (Sect. 3.1), the accuracy of OH line wavelengths was already

investigated by Cosby et al. (2006). Their theoretical line positions originate from Cosby et al. (2000) but should be similar to Goldman et al. (1998), the basis of the OH data in HITRAN, for low rotational levels. For higher rotational levels, the line wavelengths calculated by Cosby et al. (2000) should be more precise. Indeed, although the spectrum of Hanuschik (2003) is noisier, the critical OH(5-1) lines do not show clear systematic offsets. However, all OH data indicate a mean shift of the UVES-based wavelengths of about $+1.0\,\mathrm{pm}$. This offset is similar to those for atomic and molecular oxygen in the same

wavelength range, which were also measured by Cosby et al. (2006). Thus, systematic errors in the wavelength calibration are the most likely explanation. In comparison, our measurements result in a negative mean offset. This could be caused by differences in the UVES sample, data processing, and analysis.

## 4 Einstein-A coefficients

### 4.1 Full OH level populations

The intensities $I_{i'i''}$ derived in Sect. 3.2, where $i'$ and $i''$ are the upper and lower states of the roto-vibrational transition, can be converted into level populations by dividing Einstein coefficients $A_{i'i''}$. For a visualisation, these populations are usually normalised by dividing the statistical weight (i.e. the degeneracy) of the upper state $g' = g_{i'}$ and then logarithmised. Following





Noll et al. (2015), we define

$$
y := \ln\left( \frac{I_{i'i''}}{A_{i'i''}g'} \right), \tag{1}
$$

where the intensity is given in rayleighs and the Einstein-A coefficients are provided in inverse seconds, which is consistent with population column densities in units of $10^6 \, \text{cm}^{-2}$ (Noll et al., 2018b). For $\Lambda$ doublets, $i$ is characterised by the vibrational level $v$, rotational level $N$, and electronic substate $F$. In the case of individual components, the parity $p$ (i.e. e or f) is also a parameter and $g'$ is half as large as for the doublet.

Apart from the uncertainties in the line intensities, the quality of the resulting populations also depends on the reliability
of the Einstein-A coefficients. As already mentioned in Sect. 1, the latter is not satisfying as the available sets differ quite significantly. With our large sample of energy levels, where the population of each state can be derived from up to nine different lines, we can carry out a comprehensive comparison of Einstein-A coefficients. As a reference set, we take the coefficients calculated by Brooke et al. (2016) (B+16), who provide the most recent and largest set of OH line parameters. The left upper panel of Fig. 5 shows the corresponding $y$ for the 544 reliable $\Lambda$ doublets of class 3 (neglecting OH(7-1), i.e. two
doublets) as a function of the energy of the upper state $E'$. The distribution of populations displays the well-known pattern of steep population decreases for low $N'$ and weaker population gradients for higher $N'$ (Pendleton et al., 1989, 1993; Cosby and Slanger, 2007; Oliva et al., 2015; Kalogerakis et al., 2018; Noll et al., 2018b). Moreover, it indicates the expected decrease of populations for higher $v'$ with a remarkable exception for $v' = 8$ (Cosby and Slanger, 2007; Noll et al., 2015). The latter is a signature of the nascent OH level population distribution, which mainly occupies $v' = 8$ and 9. The population properties will
be discussed in more detail in Sect. 5.

It is now important to know how robust the observed pattern is with respect to changes in the set of Einstein-A coefficients. For this purpose, we also consider the HITRAN database with the version from 2012 (Rothman et al., 2013) (see also Sect. 3.3), which is mainly based on the calculations of Goldman et al. (1998) for OH. Moreover, we use the coefficients from van der Loo and Groenenboom (2008) (vdLG08), i.e. the corrected version of van der Loo and Groenenboom (2007), Turnbull and Lowe
(1989) (TL89), Langhoff et al. (1986) (LWR86), and Mies (1974) (M74). We neglect the also still popular data from Nelson et al. (1990) as their line list only focusses on low $\Delta v$ and low $N'$. Except for B+16, our selection of sets agrees with Liu et al. (2015) and Hart (2019b), who studied the impact of Einstein-A coefficients on the populations of low rotational levels. Other comparisons used a smaller number of sets (French et al., 2000; Cosby and Slanger, 2007; Noll et al., 2015; Parihar et al., 2017; Noll et al., 2018b). Note that the three oldest sets lack a significant number of the measured 723 $\Lambda$ doublets. The
set of Turnbull and Lowe (1989) only includes 634 doublets with maximum $N'$ between 13 ($R_1$ branch) and 15 ($Q_2$ and $P_2$ branches). In the case of LWR86 and M74, the $N'$-related limits are higher by 1 but the bands with $\Delta v = 6$, i.e. essentially OH(8-2) and OH(9-3), are not covered. The number of doublets is therefore only 566 or 78% of the full sample.

Figure 5 reveals clear discrepancies between the populations for the six investigated sets of Einstein-A coefficients. The general structure of the distribution is similar but the $y$ values are shifted. Taking 416 $\Lambda$ doublets with $N' \leq 12$ and $\Delta v \leq 5$,
which are present in all six sets, we find mean $y$ between $-1.43$ for TL89 and $-0.02$ for LWR86 (Table 1). This corresponds to an unsatisfyingly large population ratio of about 4.1. Substituting the extreme TL89 $y$ value by the next highest one of $-0.69$



**Figure 5.** Distribution of logarithmic OH level populations $y$, i.e. line intensity in rayleighs divided by Einstein-A coefficient in inverse seconds and the statistical weight of the upper level $g'$, as a function of the upper level energy $E'$ in inverse centimetres. The six panels show the results for the reliable $\Lambda$-doublet measurements (see Fig. 2) neglecting OH(7-1) (two lines) for different sets of Einstein-A coefficients: Brooke et al. (2016) (B+16), HITRAN (Rothman et al., 2013), van der Loo and Groenenboom (2008) (vdLG08), Turnbull and Lowe (1989) (TL89), Langhoff et al. (1986) (LWR86), and Mies (1974) (M74). The number of plotted data points is smaller for older sets due to their limitations in the coverage of faint bands and high rotational levels. The populations derived from different branches (P, Q, and R) are highlighted by different symbols and colours (see legend in lower left panel).

for HITRAN, the ratio is still about 1.9. The coefficients of M74, B+16, and vdLG08 result in intermediate mean $y$ values of $-0.33$, $-0.18$, and $-0.13$, respectively.

Any estimate of absolute OH level populations by means of OH line intensities will be highly uncertain with these results if the quality of the Einstein-A coefficients used cannot be evaluated. Tests of the accuracy of the measured absolute populations require an alternative calculation which is less sensitive to the choice of the set of molecular parameters. Using a kinetic model






**Table 1.** Comparison of populations $y$ and population ratios $\Delta y$ with respect to changes in branch and $v''$ for six sets of Einstein-A coefficients

| Set | $\langle y \rangle^{a}$ | $\langle \Delta y \rangle^{b}$ $\Delta N^{d}$ Q−P | $\langle \Delta y \rangle$ $\Delta N$ R−P | $\langle |\Delta y| \rangle^{c}$ $v''^{e}$ | $\langle |\Delta y| \rangle$ $v''$ P | $\langle |\Delta y| \rangle$ $v''$ Q | $\langle |\Delta y| \rangle$ $v''$ R |
|---|---|---|---|---|---|---|---|
| B+16 | −0.18 | −0.25 | −0.08 | 0.11 | 0.08 | 0.17 | 0.10 |
| HITRAN | −0.69 | −0.31 | −0.28 | 0.19 | 0.12 | 0.27 | 0.27 |
| vdLG08 | −0.13 | −0.25 | −0.06 | 0.17 | 0.13 | 0.23 | 0.19 |
| TL89 | −1.43 | −0.37 | −0.51 | 0.38 | 0.28 | 0.44 | 0.53 |
| LWR86 | −0.02 | −0.25 | −0.02 | 0.18 | 0.14 | 0.25 | 0.20 |
| M74 | −0.33 | −0.27 | −0.24 | 0.41 | 0.42 | 0.36 | 0.43 |
| $N_{\text{sel}}^{f}$ | 416 | 82 | 96 | 127 | 65 | 30 | 32 |

<sup>a</sup> mean logarithmic level population $y$

<sup>b</sup> mean difference of logarithmic level populations

<sup>c</sup> mean absolute difference of logarithmic level populations

<sup>d</sup> change in branch (Q−P and R−P)

<sup>e</sup> change in $v''$ for all and individual branches (P, Q, R)

<sup>f</sup> number of selected Λ doublets (present in all sets of Einstein-A coefficients)

for chemical OH production and excitation relaxation via collisions and radiative transitions is a solution, although various required rate coefficients and molecular abundances (especially for atomic oxygen) are quite uncertain. Noll et al. (2018b) found for $v = 9$ based on OH line intensities from UVES and pressure, temperature, and molecular abundance profiles from the satellite-based Sounding of the Atmosphere using Broadband Emission Radiometry (SABER) instrument (Russell et al., 1999) and the empirical atmospheric NRLMISE-00 model (Picone et al., 2002) that the higher populations related to B+16 tend to be more reliable than the lower ones based on HITRAN. Hence, at least the very low populations related to TL89 appear to be quite unlikely.

## 4.2 Detailed population comparisons

Figure 5 marks the populations derived from lines of the R, Q, and P branches by different symbols. A good set of Einstein-A coefficients should result in similar population distributions for the three branches. However, the TL89 $y$ values for the R branch are distinctly below those of the P branch. A similar but weaker effect can also be seen for the data related to HITRAN and M74. In order to study these discrepancies in more detail, we calculated differences $\Delta y$ for lines with the same upper state but different branches. We focus on Q versus P branch and R versus P branch. The corresponding results for the six sets of

**Figure 6.** Difference in logarithmic OH level populations $\Delta y$ by the change of the branch of the measured transitions from P to Q or R as a function of the upper state energy $E'$ relative to the $v'$-specific zero point (corresponding to $N' = 1$ and $F' = 1$) in inverse centimetres for six sets of Einstein-A coefficients. For more details, see legend and Fig. 5.

Einstein-A coefficients and reliable $\Lambda$ doublets of class 3 are shown as a function of $E'$ relative to the lowest energy for a given $v'$ in Fig. 6. For B+16 Einstein-A coefficients, 219 population ratios $\Delta y$ are plotted.

All sets indicate unsatisfying ratios for the comparison of Q- and P-related populations, especially for high $\Delta E'$ or $N'$, where $\Delta y$ can be lower than $-1$. The mean $\Delta y$ for a subsample with $N' \leq 12$ and $\Delta v \leq 5$ (c.f. Sect. 4.1) are between $-0.37$ for TL89 and $-0.25$ for B+16, vdLG08, and LWR86 (Table 1), i.e. the different sets fail in a similar way. According to the theoretical considerations of Pendleton and Taylor (2002) triggered by the OH(6-2) line intensity ratios measured by French et al. (2000), this can be explained by the general neglection of orbital angular momentum uncoupling, which is related to rotational-electronic mixing of the electronic ground state $X^2\Pi$ and the first excited state $A^2\Sigma^+$, for the calculation of the available Einstein-A coefficients. OH line measurements in near-infrared spectroscopic data from the Nordic Optical Telescope





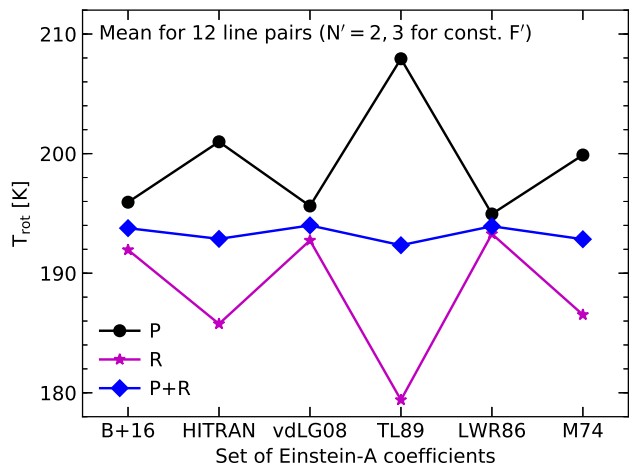

**Figure 7.** Rotational temperature $T_{\rm rot}$ in kelvins for six sets of Einstein-A coefficients (see Fig. 5). The plotted mean temperatures were derived from 12 pairs of OH lines with fixed electronic substate $F'$ originating from the second and third rotational upper level $N'$. The branch was either P (circles) or R (stars). The diamonds show the resulting $T_{\rm rot}$ for a combination of the $y(E')$ regression slopes for both branches (mean of inverse $T_{\rm rot}$).

at La Palma (Spain) by Franzen et al. (2019) indicate that too low Q-branch populations or too high Einstein-A coefficients
(based on HITRAN) are also an issue for OH bands with $\Delta v = 2$ and 3 not covered by our study.

The comparison of populations based on R- and P-branch lines reveals a more complex situation than for the Q-branch data.
All sets of Einstein-A coefficients show negative mean $\Delta y$, i.e. lower R-related populations on average (Table 1). However,
the range is relatively wide with values between $-0.51$ for TL89 and $-0.02$ for LWR86. The latter is the only satisfying set
for this comparison at low $\Delta E'$ as it was already found by French et al. (2000) for OH(6-2) low-$N'$ lines. For levels with high
rotational energy, $\Delta y$ tends to be positive. For the other sets, Fig. 6 shows a clear decrease of $\Delta y$ with increasing $\Delta E'$ at least
below $1{,}000\,{\rm cm}^{-1}$. For example, the most recent set B+16 shows mean $\Delta y$ of $-0.04$ and $-0.14$ below and above $400\,{\rm cm}^{-1}$,
respectively. At high $\Delta E'$, the negative trend appears to vanish for B+16, HITRAN, and vdLG08, the latter even indicating an
increase at the highest energies as in the case of LWR86. The especially bad performance of the TL89, HITRAN, and M74
coefficients is probably related to an underestimation of the vibration–rotation interaction (Pendleton and Taylor, 2002).
The differences in the dependence of the P- and R-branch-based populations on $E'$ for the investigated sets of Einstein-A
coefficients imply deviations in the related rotational temperatures

$$T_{\rm rot} = -\frac{1}{k_{\rm B} \frac{{\rm d}y}{{\rm d}E'}} \qquad (2)$$

(Mies, 1974; Noll et al., 2018b), where $k_{\rm B}$ is the Boltzmann constant and ${\rm d}y/{\rm d}E'$ represents the slope of a regression line in a
$y(E')$ plot like Fig. 5 for the included level populations. For a quantitative $T_{\rm rot}$ comparison, we considered pairs of levels with
a difference in $N'$ of 1 where the populations were derived from reliable lines (class 3) of the same OH band, $F'$, and branch.
Only those pairs were selected that are available for the P and the R branch. This resulted in 35 pairs that are covered by all sets


of Einstein-A coefficients up to $N' = 12$. The sample is relatively small since R-branch lines are often blended. The highest number of 12 pairs is found for the combination of the lowest $N'$ of 2 and 3 ($N' = 1$ does not exist for the R branch). Mean results for these 12 pairs (which minimise the measurement uncertainties) are shown in Fig. 7. The $T_{\mathrm{rot}}$ differences based on higher $N'$ agree qualitatively.

For the P branch, Fig. 7 reveals a wide range of mean temperatures between 195 K for LWR86 and 208 K for TL89, i.e. the selection of the set of Einstein-A coefficients strongly affects the derivation of absolute $T_{\mathrm{rot}}$. The situation is better if the extremely high value for TL89 is neglected. In this case, the maximum difference (now limited by the HITRAN-related result) is only 6 instead of 13 K. Moreover, the $T_{\mathrm{rot}}$ for the two most recent sets B+16 and vdLG08 agree well with the minimum related to LWR86. The temperature differences are consistent with those derived by Liu et al. (2015) for low-$N'$ $P_1$-branch lines of the OH bands (3-0), (5-1), (6-2), (8-3), and (9-4) based on observations with a Czerny-Turner spectrometer at Xinglong in China. The differences between the highest and lowest $T_{\mathrm{rot}}$ related to TL89 and LWR86, respectively (B+16 was not published yet), were between 9 K for OH(3-0) and 17 K for OH(8-3) with the same mean of 13 K. The trend of decreasing $T_{\mathrm{rot}}$ differences for OH bands with longer central wavelengths can also be observed in our data. Taking the differences between HITRAN and B+16 as an example, we find between 2 K for OH(3-0)$P_1$ and 9 K for OH(6-1)$P_2$ for the 12 selected line combinations. In this context, the result of Hart (2019b) for the $P_1$-branch of OH(4-2) is interesting. Based on data from an astronomical spectrograph at Apache Point in the USA, he found a maximum difference of 3 K for the same five sets investigated by Liu et al. (2015). If the minimum related to LWR86 is excluded, the variation is only about 1 K with the lowest $T_{\mathrm{rot}}$ related to TL89.

Figure 7 also shows $T_{\mathrm{rot}}$ based on R-branch lines, which were not used in the discussed studies. The set-dependent results are remarkable since they mirror those for the P-branch lines. Now, $T_{\mathrm{rot}}$ ranges from 179 K for TL89 to 193 K for LWR86, i.e. the maximum difference of 14 K is very similar to the result for the P branch but the sign is reversed. Moreover, all $T_{\mathrm{rot}}$ related to the R branch are lower than those related to the P branch. Hence, the $T_{\mathrm{rot}}$ difference between P and R branch is between 2 K for LWR86 and 29 K for TL89. For individual double pairs of lines, R-branch-related $T_{\mathrm{rot}}$ can also be higher than those for the P branch, i.e. LWR86 might not show the smallest differences. However, the large discrepancies for TL89 are obvious in any case.

As the P- and R-branch $T_{\mathrm{rot}}$ show an oppositional behaviour, we averaged the slopes $\mathrm{d}y/\mathrm{d}E'$ for both branches to derive more robust temperatures. As demonstrated by Fig. 7, this was achieved. The mean value for all sets is 193.3 K with a standard deviation of only 1.0 K. The latter represents less than 20% of the variation for the individual branches. Consequently, the combination of P- and R-branch data can significantly reduce the impact of the choice of the Einstein-A coefficients on the quality of the resulting $T_{\mathrm{rot}}$. However, in practice, this will be difficult to apply due to the difficulties in measuring R-branch lines at moderate spectral resolution. Hence, it is more promising to improve the Einstein-A coefficients by a better handling of the vibration–rotation interaction (Pendleton and Taylor, 2002), which appears to be the main reason for the set-dependent $T_{\mathrm{rot}}$ discrepancies. Data as plotted in Fig. 7 can provide important constraints for this purpose.

Another population-independent evaluation of Einstein-A coefficients is possible for transitions with the same upper and lower levels except for a different $v''$. For the comparison of the related $y$, it was necessary to define a reference OH band for each $v'$ between 4 and 9, where we have line measurements for two or more bands. We preferentially selected bands with good

**Figure 8.** Difference in logarithmic OH level populations $\Delta y$ by the change of the lower vibrational level $v''$ of the measured transitions as a function of the mean wavelength of the $\Lambda$ doublet in micrometres for six sets of Einstein-A coefficients. For the reference bands OH(4-0), OH(5-1), OH(6-2), OH(7-3), OH(8-3), and OH(9-4), the resulting $\Delta y$ are zero and are therefore not shown. OH(3-0) is also neglected since it is the only band with $v' = 3$. Finally, additional bands at short wavelengths are missing in the case of the limited sets of Langhoff et al. (1986) and Mies (1974). For more details, see legend and Fig. 5.

quality data in the middle of the covered wavelength range: OH(4-0), OH(5-1), OH(6-2), OH(7-3), OH(8-3), and OH(9-4). The resulting $\Delta y$ are plotted in Fig. 8 as a function of line wavelength for the six sets of Einstein-A coefficients. In the case of B+16, population ratios for 182 pairs of reliable $\Lambda$ doublets (class 3) are shown. For LWR86 and M74, this number is only 136 365 due to the limitations in $N'$ and $\Delta v$ (Sect. 4.1). The plots indicate a complex behaviour where the $\Delta y$ depend on band, branch, and $N'$ in a different way for each set of Einstein-A coefficients. The data points for the lowest $N'$, which tend to cluster for each band, show a clear trend with wavelength (or $\Delta v$) for all sets but B+16. The data for HITRAN, vdLG08, and TL89





indicate an increase of $\Delta y$ with wavelength, whereas the M74 data show a decrease. For LWR86, $\Delta y$ is mainly negative, i.e. the reference bands in the middle of the wavelength range with $\Delta y = 0$ (not plotted) indicate the highest relative populations.

The overall performance of each set can be evaluated by measuring the mean absolute $\Delta y$ for line pairs where Einstein-A coefficients are available in all sets. The corresponding results for 127 line pairs fulfilling $\Delta v \leq 5$ and $N' \leq 12$ are provided in Table 1. The highest and hence worst $\langle|\Delta y|\rangle$ were found for M74 (0.41) and TL89 (0.38). Lower but still unsatisfactory values of around 0.18 were obtained for HITRAN, vdLG08, and LWR86. B+16 clearly shows the best performance with a value of 0.11. Table 1 also contains $\langle|\Delta y|\rangle$ depending on branch. The best results are obtained for the P branch for all sets but M74,
which is unsatisfying for all branches.

The large $\Delta y$ and their trend with wavelength for M74 and TL89 shown in Fig. 8 were already found by Cosby and Slanger (2007). Also using UVES data, they compared the populations derived from the $P_1(1)$ line of the accessible OH bands with $v'$ of 6, 8 and 9. Including the transition probabilities of M74, LWR86, TL89, and Goldman et al. (1998), their analysis favoured the latter, i.e. the main input source for HITRAN. Cosby and Slanger (2007) explained the bad performance of the TL89
coefficients by the erroneous intensity calibration of data used for the applied empirical dipole moment function (DMF), which is the basis for the calculation of the transition probabilities. In general, it can be expected that the accuracy of Einstein-A coefficients for bands with high $v'$ in the optical tends to be worse than in the case of bands with low $v'$ in the near-infrared, which are not accessible by UVES. Theoretical ab initio DMF calculations as used by Mies (1974) and van der Loo and Groenenboom (2007, 2008) are more uncertain for internuclear distances between the O and H atom far from the equilibrium.
Moreover, the input data for empirical DMFs (Turnbull and Lowe, 1988, 1989; Nelson et al., 1990), theoretically extended empirical DMFs (Goldman et al., 1998), and modified ab initio DMFs (Langhoff et al., 1986; Brooke et al., 2016) were mainly restricted to low $v$ or low $\Delta v$.

As discussed in Sect. 3.2, about half the measured $\Lambda$ doublets are resolved due to the high spectral resolving power of UVES. This allowed us to systematically study deviations between the Einstein-A coefficients of the e and f components. The
older sets of transition probabilities (M74, LWR86, and TL89) do not provide information on the individual components. The HITRAN database (Gordon et al., 2017) contains these components but the Einstein-A coefficients were just set to the value of the corresponding doublet. Finally, vdLG08 and B+16 consider $\Lambda$ doubling but the differences between the coefficients are very small. For B+16, the mean relative difference for our sample of 723 doublets is only 0.04%. The largest deviations are related to P- and Q-branch lines with high $N'$. The maximum in our sample of 0.25% is linked to OH(5-1)$Q_2$(6). As the
corresponding values for vdLG08 are almost identical, it is sufficient to use only B+16 coefficients for the comparison of the $\Lambda$-doublet components.

Figure 9 shows the results for 185 reliable (class 3) doublets with resolved components. The logarithmic population ratio $\Delta y$ for f minus e for the upper state parity $p'$ is plotted as a function of the corresponding difference in $E'$. The latter is negative for $F' = 2$ lines of the P and R branch according to the parity definition used by Brooke et al. (2016). If the theoretically predicted
equality of the transition probabilities was true, $\Delta y$ should be close to 0. Small deviations in the populations are possible due to the small differences in $E'$. Assuming a Boltzmann-like distribution for a typical kinetic temperature of 190 K at altitudes of the OH emission layer at Cerro Paranal (Noll et al., 2016), there would be $\Delta y = +0.08$ for $\Delta E' = -10\,\mathrm{cm}^{-1}$ and the





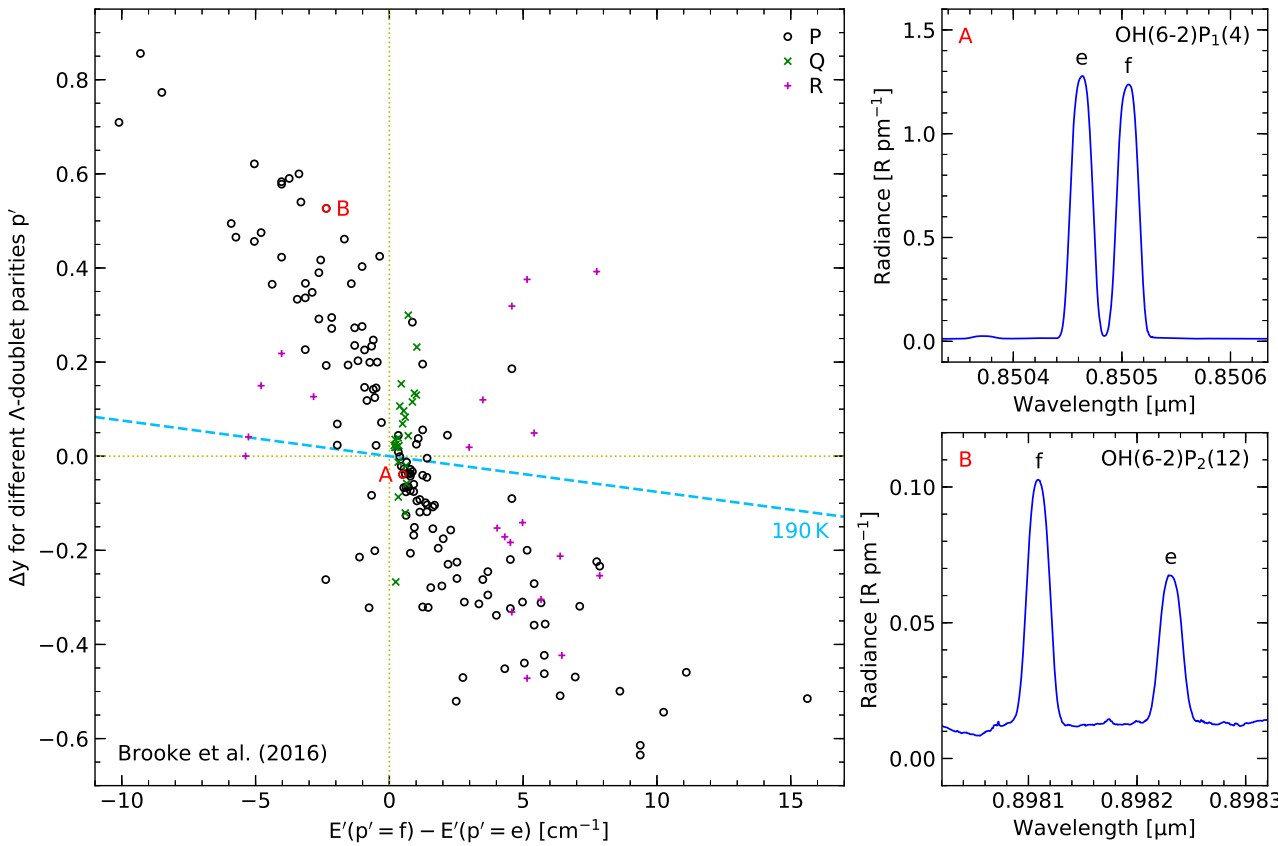

**Figure 9.** Difference in logarithmic OH level populations $\Delta y$ of the separately measured components of reliable $\Lambda$ doublets indicated by the upper level parity f and e as a function of the corresponding difference in the upper level energy $E'$. The results for different branches (P, Q, and R) are marked by different symbols and colours (see legend). Examples for small and large population discrepancies are marked by the letters A and B. On the right-hand side, the corresponding $\Lambda$-doublet spectra are plotted with indicated line identifications. The dashed line in the main plot indicates the effect of a thermal population with a temperature of 190 K on $\Delta y$.

same amount with negative sign for the corresponding positive energy difference. However, the true $\Delta y$ are about one order of magnitude larger. The average absolute discrepancy in $\Delta y$ is 0.22. In the case of large energy differences of at least $5\,\mathrm{cm}^{-1}$,

it would even be 0.37, which corresponds to a ratio of 1.4. The clear differences in the strengths of the $\Lambda$-doublet components are also illustrated by two example spectra. The weakly separated OH(6-2)P$_1$(4) doublet already shows slight differences in the intensity ($\Delta y = -0.04$). For the widely separated OH(6-2)P$_2$(12) pair, the f component is about 1.7 times brighter than the e component ($\Delta y = 0.53$). It is astonishing that it appears that these large effects have not been recognised, so far. They can only be explained by inadequate Einstein-A coefficients since the P, Q, and R branches behave differently. The $\Delta y$ related to

P-branch lines could be fitted by a non-LTE Boltzmann-like distribution with about 20 K. However, the $\Delta y$ distributions for the Q and R branches are less clear. A convincing regression line cannot be drawn, and even if the fitting was performed, the

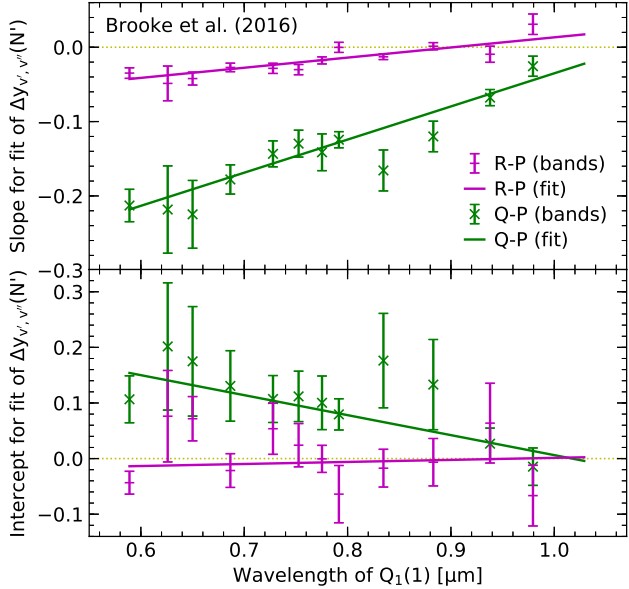

**Figure 10.** Fit of the differences in logarithmic OH level populations for Q (crosses) and R branches (horizonzal bars) relative to the P branch as a function of the upper rotational level $N'$ for each band where a reliable linear regression could be performed. The symbols with error bars show the resulting slope (upper panel) and intercept (lower panel) and their respective uncertainties. These values based on Einstein-A coefficients of Brooke et al. (2016) were also fitted by performing error-weighted linear fits depending on the wavelength of $Q_1(1)$ for each band. The fit lines are displayed.

slopes would be very different. This rules out a significant impact of possible non-LTE-inducing propensity differences for the population of the e and f states, even if the parity definitions are changed or more complex dependencies involving several level parameters are considered.

## 4.3 Correction of Brooke et al. coefficients

The discussion in the previous section has shown that the currently available sets of Einstein-A coefficients are not satisfying, especially with respect to Q-branch lines and Λ-doublet components. Overall, the B+16 set is the most promising since it is the most complete in terms of the included lines, shows the smallest band-to-band variations for constant $v'$, and is only slightly worse than LWR86 with respect to the population deviations between different branches. Hence, we focus on the
B+16 set for the rest of this paper. However, as the remaining issues can still negatively affect the evaluation of OH level population distributions as shown in Fig. 5, we tried to improve the coefficients to result in more consistent population ratios in the diagnostic plots discussed in Sect. 4.2. Our approach is fully empirical, i.e. it is based on regression lines and correction factors, and consists of three steps related to the correction of the discrepancies revealed by Figs. 6, 8, and 9. Complex fitting approaches involving theoretical DMF-based calculations of the Einstein-A coefficients are out of the scope of this study.





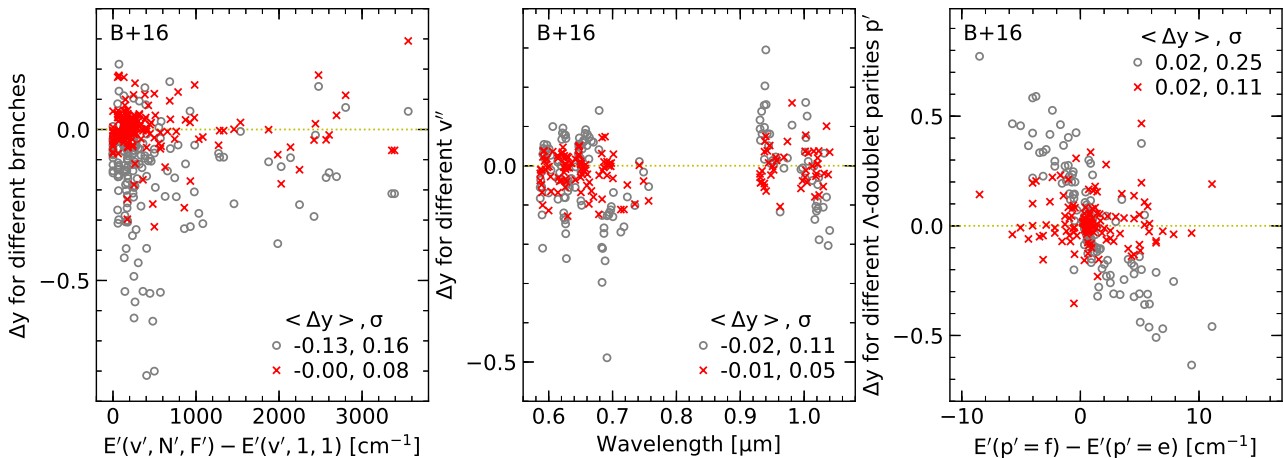

**Figure 11.** Correction of Brooke et al. (2016) Einstein-A coefficients. The measured (circles) and corrected (crosses) OH level population ratios $\Delta y$ are given for changes in the branch (left; c.f. Fig. 6), lower vibrational level $v''$ (middle; c.f. Fig. 8), and the $\Lambda$-doublet component (right; c.f. Fig. 9). Each plot lists the mean $\Delta y$ and standard deviation $\sigma$ for the measured and corrected data.

We started with a correction of the discrepancies between the populations derived from different branches as shown in Fig. 6. For this purpose, we used 184 suitable, highly reliable $\Lambda$ doublets classified as 33, i.e. with a primary and secondary class of 3 (Sect. 3.2). It turned out that a linear fit of $\Delta y$ for Q versus P and R versus P shows the best performance for $N'$ as the independent variable (instead of the plotted $\Delta E'$) and a separate fit for each OH band. The resulting slopes and intercepts and their uncertainties are provided in Fig. 10. There are no data points for OH(5-0), OH(9-5), and OH(4-1) due to relatively high

uncertainties caused by an insufficient number of reliable line measurements. For the remaining 12 bands, 5 to 16 (3 to 8) $\Lambda$ doublets could be used for the fit of the difference between R and P branch (Q and P branch). The slopes for the Q branch are clearly more negative than those for the R branch. The mean values are $-0.15$ and $-0.02$, i.e. the apparent populations decrease by 14 and 2% compared to the P branch for an increase of $N'$ by 1. The corresponding mean intercepts are $+0.11$ and $+0.01$, i.e. nearly zero for R versus P. The data points are plotted as a function of the central wavelength of the $Q_1(1)$ doublet of each

band as especially the slope indicates a significant trend with wavelength, which suggests that the branch-related errors in the B+16 Einstein-A coefficients depend on the energy difference between the upper and lower state. The discrepancies between the branches appear to be smaller for longer wavelengths and might even vanish around 1 $\mu$m. We used these correlations for a more robust correction approach involving error-weighted linear fits of slope and intercept as a function of $Q_1(1)$ wavelength. The error weighting allowed us to consider the strong dependence of the uncertainties on the band. The resulting fits are also

shown in Fig. 10. The slopes change with $+0.45\pm0.05$ (Q) and $+0.14\pm0.03$ (R) per micrometre. For the intercepts, we obtain changes of $-0.36\pm0.08$ (Q) and $+0.04\pm0.11$ (R) per micrometre.

According to Eq. (1), the reciprocals of the population ratios $\Delta y$ from our two-step fitting procedure indicate the systematic deviations of the Einstein-A coefficients. Thanks to the derived regression lines, they can also be predicted for missing lines and bands, at least in the covered wavelength range. For the correction, we need to define a reference. While the Q-branch





coefficients do not appear to be reliable in general, it would be arbitrary to choose the P or the R branch. However, Fig. 7 suggests that the errors in the Einstein-A coefficients are minimised if both branches are combined with the same weight. Hence, we corrected the transition probabilities for the three branches by using half the fitted deviation between P and R branch as the reference. Figure 11 (left) shows $\Delta y$ for the 184 considered pairs of $\Lambda$ doublets before and after the correction. While in the former case the mean value and standard deviation are $-0.13$ and $0.16$, the corrected data reveal $0.00$ and $0.08$.

Thus, the offsets vanished completely on average and the scatter was reduced by a factor of 2.

In the next step, we corrected $\Delta y$ offsets between lines differing only in $v''$ as shown in Fig. 8. For this purpose, we focussed on relatively bright lines with $N' \leq 4$ for the P and R branch and $N' = 1$ for the Q branch. As these $\Lambda$ doublets indicate similar $\Delta y$ (scatter of 0.03), differences in the selected line subsets do not critically affect the mean values. For each considered OH band, 8 to 14 reliable pairs of $\Lambda$ doublets of class 33 were available (89 in total). For the correction of $\Delta y$ by changing the

Einstein-A coefficients, we used the same reference bands for each $v'$ as discussed in Sect. 4.2. The choice is motivated by the accessibility and quality of the line measurements with UVES. It does not necessarily include the bands with the most realistic transition probabilities. As discussed in Sect. 4.2, coefficients of bands with low $v'$ and $\Delta v$ tend to be more reliable as the DMF calculations are less challenging and the experimental data are more abundant. Our reference bands have $v'$ between 3 and 9 and $\Delta v$ between 3 and 5. Hence, the most promising bands are beyond the UVES wavelength range. Nevertheless, there does

not appear to be a strong quality gradient with $\Delta v$ or wavelength ($v'$ cannot be tested) since the B+16 coefficients do not show such a dependence of $\Delta y$ for the covered bands in Fig. 8 (in contrast to the other investigated sets). Note that this is different from the situation for the branches illustrated in Fig. 10. In the end, we shifted the mean $\Delta y$ for eight OH bands to zero by multiplying the Einstein-A coefficients by factors between 0.85 for OH(7-2) and 1.06 for OH(8-4). This reduces the scatter in the measured populations for fixed $v'$ in any case, even if the choice of the reference bands might not be optimal. Figure 11

(middle) shows the corresponding results for 143 $\Lambda$ doublets. The discussed corrections (also including the branch-related modifications) change the mean $\Delta y$ and standard deviation from $-0.02$ and $0.11$ to $-0.01$ and $0.05$.

Finally, we corrected the $\Delta y$ between the $\Lambda$-doublet components as shown in Fig. 9. This is necessary in order to also use doublets with only one reliable component for the study of the OH level populations discussed in Sect. 5. For the change of the Einstein-A coefficients, we assumed a natural population discrepancy between the two components consistent with a

temperature of 190 K as illustrated in Fig. 9. We fitted the remaining $\Delta y$ for each branch using $N'$ as the independent variable due to a better performance with respect to linear regressions compared to $\Delta E'$. This approach requires to flip the sign for the data points with negative $\Delta E'$. This is reasonable since the amount of the deviations is very similar for $\Lambda$ doublets with $F'$ of 1 and 2. For the fits related to P, Q, and R, we considered 99, 18, and 11 resolved doublets of class 33. The resulting slopes (and intercepts) are $-0.035 \pm 0.003$ ($+0.13 \pm 0.03$), $+0.065 \pm 0.019$ ($-0.19 \pm 0.08$), and $-0.035 \pm 0.035$ ($+0.43 \pm 0.48$),

respectively. Hence, the effect for the Q branch seems to be twice as large as for the P branch and to also have a different sign. The R branch might behave similarly to the P branch but the uncertainties are high. Assuming that the e and f components equally contribute to the fitted differences, we corrected the Einstein-A coefficients for the reliable $N'$ range of the fits, i.e. we neglected doublets with low $N'$ where both components are not sufficiently separated or the fit crossed $\Delta y = 0$. Figure 11





(right) shows the resulting change in the mean $\Delta y$ and scatter for the investigated 128 doublets. While the small mean value of
$+0.02$ did not significantly change, the standard deviation was clearly reduced from 0.25 to 0.11.

## 5  OH level populations

### 5.1  Mean populations and rotational temperatures

With the correction of the B+16 Einstein-A coefficients in Sect. 4.3, we minimise the scatter in the OH level populations for
upper states with different measured lines. Moreover, the change of the populations with $N'$ appears to be more reliable due
to the promising combination of P- and R-branch data for $T_{\mathrm{rot}}$ estimates (Fig. 7). The resulting population distributions for $v'$
between 4 and 9 are shown in Fig. 12. We neglect $v' = 3$ due to the lack of high $N'$ states in the UVES data (Fig. 3), which
does not allow us to describe this population distribution in detail. As already briefly discussed in Sects. 1 and 4.1, there is a
characteristic pattern for increasing level energy with a steep population decrease for low $N'$ and a rather slow decrease for
high $N'$. The difference between high and low $N'$ tends to increase with decreasing $v'$. Moreover, the change between the two
extremes does not appear to happen continuously with increasing $N'$. Instead, the transition is mostly localised in a narrow
$\Delta E'$ interval of a few hundred inverse centimetres. This known pattern (Cosby and Slanger, 2007; Oliva et al., 2015) suggests
the definition of a cold and a hot population for each $v'$, which can be described by corresponding $T_{\mathrm{rot}}$ and population ratios
for fixed level energies (Oliva et al., 2015; Kalogerakis et al., 2018; Kalogerakis, 2019).

We applied this concept by fitting the natural logarithm of the sum of two exponential Boltzmann terms as a function of
$\Delta E'$ ($E'$ relative to the energy for $N' = 1$ and $F' = 1$) to the corrected $y$ for each $v'$. We considered the populations from
all $\Lambda$ doublets with quality classes above 0. For classes 1 and 2, we derived the doublet-related populations from the reliable
components. An inspection of the change of the populations with increasing $\Delta E'$ resulted in the rejection of the highest $N'$
levels for $v' = 8$ and 9 as the related populations cannot satisfyingly be reproduced by a two-component fit. In the case of
$v' = 9$, the seven measurements with $N' \geq 10$ were neglected. All corresponding $E'$ are between 250 and 790 $\mathrm{cm}^{-1}$ above the
exothermicity limit of the hydrogen–ozone reaction (Sect. 3.2), which can explain the rapid population decrease in this energy
range, which could not clearly be constrained before due to a lack of data (Cosby and Slanger, 2007; Noll et al., 2018b). In
the case of $v' = 8$, a strong decrease of the populations is found for eight measurements related to $N' \geq 14$ with $E'$ between
660 $\mathrm{cm}^{-1}$ below and 90 $\mathrm{cm}^{-1}$ above the exothermicity limit. This is an interesting result as it provides valuable constraints on
the nascent populations and the relaxation process from $v' = 9$ to 8. The drop of the populations below the exothermicity limit
also seems to be present in the population distribution of Cosby and Slanger (2007), also based on UVES spectra (Hanuschik,
2003; Cosby et al., 2006). However, the authors do not discuss this phenomenon. Our $v' \leq 7$ data, which are related to energies
of more than 1,200 $\mathrm{cm}^{-1}$ below the limit, do not show a cut in the populations. The data of Cosby and Slanger (2007) are not
conclusive here either.

The remaining population measurements for each $v'$, which varied between 83 for $v' = 4$ and 124 for $v' = 6$ and 8, were
fitted with our two-component model by means of robust least-squares minimisation, which resulted in the same best fits for
a wide range of start values. Figure 12 shows the final best fits and also indicates the corresponding temperatures $T_{\mathrm{cold}}$ and







**Figure 12.** Distribution of logarithmic OH level populations for the corrected Brooke et al. (2016) Einstein-A coefficients. Except for OH(7-1), all $\Lambda$ doublets with at least one reliable component are considered. The populations are plotted in separate panels for each upper vibrational level $v'$ between 4 and 9. The given level energies in inverse centimetres are relative to the lowest $v'$-related energy. The populations can be fitted by means of a two-component rotational temperature fit. All data points involved are marked by circles. The few exceptions (indicated by crosses) are close to or above the exothermicity limit of the hydrogen–ozone reaction, which is marked by vertical dotted lines. For these data, a more complex fit would be necessary. The final fit curves are displayed by solid lines. The underlying cold and hot components are shown by dashed and dot-dashed lines, respectively. Each panel lists the $v'$-specific number of selected data points $N_{\mathrm{sel}}$, the rotational temperatures of the cold and hot components $T_{\mathrm{cold}}$ and $T_{\mathrm{hot}}$ in kelvins, and the ratio $r_{\mathrm{pop,0}}$ of the populations of both linear fit components for the lowest $v'$-related energy in per cent.

$T_{\mathrm{hot}}$ as well as the ratio of the hot and cold populations for $\Delta E' = 0$, $r_{\mathrm{pop,0}}$. Under consideration of the fit uncertainties, the best-fit $T_{\mathrm{cold}}$ are very similar and consistent with a temperature of 190 K, i.e. the typical ambient temperature at OH emission altitudes (Noll et al., 2016). Only $v' = 4$ with $196 \pm 4$ K might slightly be higher. This could point to the weak influence of an





intermediate population for low $v'$. The second highest $T_{\rm cold}$ value of $193 \pm 5 \, {\rm K}$ for $v' = 5$ would be in agreement with this interpretation. Note that fixing the fit to a $T_{\rm cold}$ of 190 K did not significantly change the other parameters. The differences were much smaller than the uncertainties. In contrast to $T_{\rm cold}$, $T_{\rm hot}$ shows a strong trend with $v'$. The temperatures increase from about 700 K for $v' = 9$ to about 7,000 K for $v' = 4$. In parallel, $r_{\rm pop,0}$ decreases from about 3% for $v' = 9$ to about 0.3% for $v' = 4$, i.e. hot populations with higher $T_{\rm hot}$ show lower contributions to the total population at low $N'$. The strong change

in $r_{\rm pop,0}$ appears to be mainly caused by the decrease of the cold population with increasing $v'$ since the $\Delta E' = 0$ intercepts of the lines describing the hot populations are located at similar $y$ values of around $-2$ in Fig. 12. The fits for $v' \leq 7$ (no rejection of states) are convincing with respect to the assumption of a homogeneous hot population, which can be described by a single temperature. Nevertheless, some fine structure might exist as the comparison of the individual measurements and the fit lines suggest, although the possible population deviations appear to be not larger than 30%, which is small compared to

population changes of the order of a magnitude in the $v'$-dependent energy ranges most contributing to $T_{\rm hot}$. Hence, the two-component fits are quite robust as the listed errors show. The highest uncertainties are related to $v' = 9$ since the hot population is essentially constrained in an energy range of less than $300 \, {\rm cm}^{-1}$, which includes 12 measurements with $N'$ of 8 and 9.

      Two-component fits were previously performed by Oliva et al. (2015) based on a spectrum taken during 2 hours with the near-infrared echelle spectrograph GIANO directly pointing to the night sky at the La Palma Observatory in Spain. The spectrum

had a resolving power of 32,000 and covered the wavelength range from 0.97 to 2.4 $\mu$m. Consequently, the investigated lines belong to OH bands with low $\Delta v$ and are complementary to those covered by our study. For the calculation of the populations, Oliva et al. (2015) used the Einstein-A coefficients from van der Loo and Groenenboom (2007). For the fits, $T_{\rm cold}$ was fixed at 200 K. The resulting $T_{\rm hot}$ and $r_{\rm pop,0}$ varied from about 1,300 K and 1.8% for $v' = 8$ to about 7,000 K and 0.23% for $v' = 4$. Although errors were not reported, these values are in good agreement with our results provided in Fig. 12. The fit parameters

for $v' = 9$ are highly uncertain. However, Oliva et al. (2015) succeeded in fitting the populations for $v' = 2$ and 3, which show an extension of the trend found for the higher $v'$. For $v' = 2$, $T_{\rm hot}$ and $r_{\rm pop,0}$ resulted in 12,000 K and 0.14%, respectively. The GIANO data were refitted by Kalogerakis et al. (2018) with unconstrained $T_{\rm cold}$, which resulted in temperatures of about 190 K but with larger scatter than in our case. For $T_{\rm hot}$, the general trend was the same but with a large step from 900 K for $v' = 8$ to 4,000 K for $v' = 7$, which disagrees with our findings. Population ratios were not provided by Kalogerakis et al. (2018). Noll

et al. (2018b) already published populations related to $v' = 9$ and P-branch lines based on the UVES data used in this study and B+16 Einstein-A coefficients. Kalogerakis (2019) fitted these populations and found $T_{\rm cold}$ and $T_{\rm hot}$ of about 180 K and 500 K, respectively. Both temperatures are lower than our results, but less than two standard deviations. The fit of Kalogerakis (2019) based on fewer data points seems to be related to a higher impact of the hot population at low $N'$. Our results for $T_{\rm hot}$ allow an interesting comparison to the $T_{\rm rot}$ of the nascent populations of $v'$ between 7 and 9, which were derived by Llewellyn

and Long (1978) using laboratory data from Charters et al. (1971). Our best-fit $T_{\rm hot}$ of $690 \pm 120$, $1340 \pm 50$, and $2180 \pm 50 \, {\rm K}$ agree well with their $760 \pm 20$, $1230 \pm 30$, and $1940 \pm 200 \, {\rm K}$, which implies that the OH relaxation processes do not appear to significantly affect the hot populations of the highest $v'$.

      The previous discussion has shown that bimodality is a good concept for the description of the population distributions for each $v'$. Moreover, the derived $T_{\rm cold}$ are close to the expected effective ambient temperatures for the $v'$-dependent OH emission





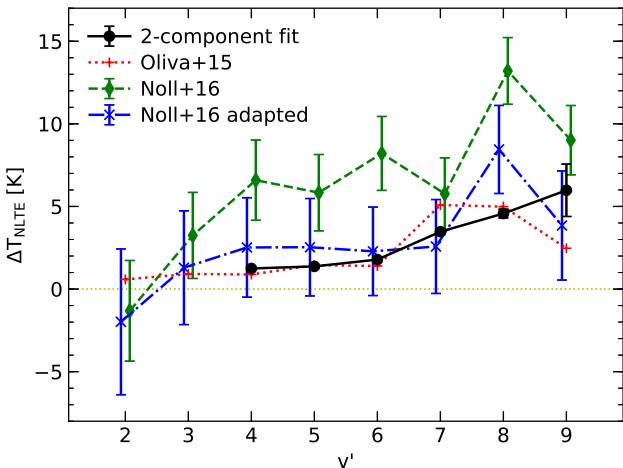

**Figure 13.** Estimate of non-LTE contributions to rotational temperatures for the three lowest rotational lines with $F' = 1$ as a function of the upper vibrational level $v'$. The differences of the $T_{\rm rot}$ for the two-component fits and the related $T_{\rm cold}$ shown in Fig. 12 are provided by circles and solid lines. Error bars (which are only larger than the symbols for $v' = 9$) are also given. Two-component fits of OH level populations were also performed by Oliva et al. (2015) based on data from the near-infrared spectrograph GIANO at La Palma Observatory in Spain. We used their fit parameters (no errors) to calculate $\Delta T_{\rm NLTE}$ for $v'$ from 2 to 9 (plus signs and dotted lines). For comparison, the figure also indicates the estimates by Noll et al. (2016) based on X-shooter spectra from Cerro Paranal and SABER data for a similar area (diamonds and dashes). As these $\Delta T_{\rm NLTE}$ were derived for Einstein-A coefficients from HITRAN, we recalculated the non-LTE effects using our corrected Brooke et al. (2016) coefficients (crosses and dot-dashed lines).

layers (Noll et al., 2016). The increasing trend of $T_{\rm rot}$ derived from the lines with the lowest $N'$ for increasing $v'$ (Cosby and Slanger, 2007; Noll et al., 2015, 2017) is not found in the best-fit $T_{\rm cold}$. Hence, our fits could be used to estimate the non-LTE contributions to such $T_{\rm rot}$, $\Delta T_{\rm NLTE}$, which are an issue for the use of $T_{\rm rot}$ as indicators of the temperatures in the mesopause region. Kalogerakis et al. (2018) and Kalogerakis (2019) compared $T_{\rm cold}$ fits with $T_{\rm rot}$ from linear regressions for levels with $\Delta E'$ lower than 500 and $250\,{\rm cm^{-1}}$, respectively. The results indicate higher $T_{\rm rot}$ than $T_{\rm cold}$ at least for the highest $v'$ (order

of 20 K). However, the uncertainties are large due to the strong impact of the line selection (Noll et al., 2015), uncertainties in the line intensities (unclear for the GIANO data), and the choice of the Einstein-A coefficients (Fig. 7). Hence, we applied a different approach by directly taking the two-component fit for the measurement of $T_{\rm rot}$. For this purpose, we derived the populations related to the first three $P_1$-branch lines, which are often taken for $T_{\rm rot}$ determinations (e.g. Schmidt et al., 2013; Noll et al., 2016), from the fit curve at the corresponding $v'$-dependent $\Delta E'$. The related $T_{\rm rot}$ were then calculated by a linear

regression of the three $y$ for each $v'$. Finally, the resulting $\Delta T_{\rm NLTE}$ is just the difference between $T_{\rm rot}$ and $T_{\rm cold}$. This method is very robust as it is fully based on the two-component fit, which relies on a high number of population measurements. Hence, uncertainties related to individual lines are negligible.

  Our $\Delta T_{\rm NLTE}$ for $v'$ between 4 and 9 are shown in Fig. 13. They increase relatively slowly between 4 and 6 from $1.2 \pm 0.1\,{\rm K}$ to $1.8 \pm 0.1\,{\rm K}$ and then faster to $6.0 \pm 1.6\,{\rm K}$ for $v' = 9$. As the latter value is relatively uncertain, $4.6 \pm 0.3\,{\rm K}$ for $v' = 8$ might also





be the maximum deviation. The errors were derived by displacing the hot component fit in both $y$ directions according to the
uncertainty in $r_{\mathrm{pop},0}$ and refitting $T_{\mathrm{hot}}$ as the only parameter to obtain modified $\Delta T_{\mathrm{NLTE}}$. This approach considers that $r_{\mathrm{pop},0}$
and $T_{\mathrm{hot}}$ are anticorrelated and that the relative uncertainty in $T_{\mathrm{cold}}$ is relatively small. Additional systematic uncertainties are
caused by assuming only two components. It is required that the fit line for the hot component can be linearly extrapolated
to $\Delta E' = 0$. The good quality of the fits in the transition region between the dominance of the cold and hot components are
promising. Nevertheless, contributions of additional components of intermediate temperature cannot be excluded (as e.g. for
$v' = 4$ due a possibly elevated $T_{\mathrm{cold}}$). Fits with such an additional component and a fixed $T_{\mathrm{cold}}$ of 190 K, where the best-fit
parameters of the intermediate and hot populations are not well constrained, showed possible $\Delta T_{\mathrm{NLTE}}$ increases by 10 to 30%,
i.e. the significance of positive non-LTE effects for all $v'$ would remain high. It is hard to imagine situations where $\Delta T_{\mathrm{NLTE}}$
could significantly drop. A sharp cut of the hot population for low $N'$ would be inconsistent with a Boltzmann-like distribution
as expected for relaxation processes. Another source of possible systematic errors are the Einstein-A coefficients, especially
with respect to their dependence on $N'$. The latter was changed in Sect. 4.3 for the B+16 coefficients by the branch-specific
corrections. Hence, we tested what happens if we consider the P- or R-branch data as the standards instead of a combination
of both. These modifications would change directly measured $T_{\mathrm{rot}}$ to values as indicated in Fig. 7. However, the effect on the
two-component approach is much smaller. It is of the order of the already small fit errors for all $v'$. As expected, non-LTE
contributions related to R as the standard are lower than those for P. Furthermore, we investigated the influence of the choice of
the energy levels on $\Delta T_{\mathrm{NLTE}}$ by also simulating line sets consisting of the first two and first four $P_1$-branch lines. For $v' = 8$ as
an example, these changes cause $\Delta T_{\mathrm{NLTE}}$ of 3.3 and 6.9 K, which clearly deviate from the plotted 4.6 K. Hence, the non-LTE
contributions are very sensitive to the selected energy levels, which is consistent with the results from Noll et al. (2015, 2018b).

As shown in Fig. 13, we also calculated $\Delta T_{\mathrm{NLTE}}$ from the two-component fits of Oliva et al. (2015). Excluding their very
uncertain fit parameters for $v' = 9$, there is a very good agreement with differences smaller than 0.4 K. The only exception
is $v' = 7$, where our non-LTE contributions are about 1.6 K lower. As the data basis and analysis were completely different
(including different Einstein-A coefficients), this convincing result demonstrates the robustness of the approach. The GIANO-
related data for $v' = 2$ and 3 suggest that $\Delta T_{\mathrm{NLTE}}$ decreases only very slowly with decreasing $v'$. The drop in $r_{\mathrm{pop},0}$ seems to
be nearly compensated by the increase in $T_{\mathrm{hot}}$.

Mean $\Delta T_{\mathrm{NLTE}}$ for $v'$ from 2 to 9 at Cerro Paranal were already derived by Noll et al. (2016) based on measurements of
25 OH bands and two $O_2$ bands (where non-LTE effects are less important) in optical and near-infrared spectra from the
echelle spectrograph X-shooter as well as from OH emission and kinetic temperature profile measurements with SABER.
The $\Delta T_{\mathrm{NLTE}}$ from the complex analysis for the first three $P_1$-branch lines (derived from different band-specific line sets) are
shown in Fig. 13. The values with a conspicuous maximum of $13.2 \pm 2.0$ K at $v' = 8$ are clearly higher than those from the
two-component fit. However, Noll et al. (2016) used HITRAN Einstein-A coefficients, which significantly deviate from our
modified B+16 coefficients. As demonstrated by Fig. 7, the impact on $T_{\mathrm{rot}}$ can be large. Hence, we recalculated the $\Delta T_{\mathrm{NLTE}}$
of Noll et al. (2016) with the modified B+16 transition probabilities for the lines considered in Sect. 4.3 and the original ones
(which result in about 2 K higher $T_{\mathrm{rot}}$ on average) in all other cases. Figure 13 indicates a clear reduction of $\Delta T_{\mathrm{NLTE}}$ for the
relevant $v' \geq 4$, which better matches our results based on two-component population fits. Between $v'$ of 4 and 7 the non-LTE





contributions are almost constant with a mean of 2.5 K. However, the absolute uncertainties are larger. Only the maximum of $8.4 \pm 2.7\,\mathrm{K}$ at $v' = 8$ seems to be significant. It might also be present (but less pronounced) in the population fitting results. The high absolute uncertainties from temperature comparisons ($v'$-related differences are safer) are a critical drawback of that method and imply that two-component population fits provide the best constraints for $\Delta T_{\mathrm{NLTE}}$, so far. The higher errors compared to the original Noll et al. (2016) data are partly related to the unavoidable mixture of corrected and uncorrected B+16

coefficients. However, B+16 line parameters also appear to cause a larger scatter in $T_{\mathrm{rot}}$ for different bands with the same $v'$ compared to HITRAN data. The change in the $\Delta T_{\mathrm{NLTE}}$ differences between adjacent $v'$ (especially around $v' = 7$) is mainly caused by a different calculation of $T_{\mathrm{rot}}$ for the reference line set consisting of the first three $P_1$-branch lines. Instead of using a constant temperature offset for the conversion from the reference line set of Noll et al. (2015) including all P-branch lines up to $N' = 3$ as discussed by Noll et al. (2016), we directly corrected the band-specific $T_{\mathrm{rot}}$ to be representative of the more

recent reference line set.

### 5.2 Population variability

The discussion of the roto-vibrational level populations of OH in Sect. 5.1 was only based on line intensity measurements in a single mean spectrum. We can learn more about these populations if we also consider variations in the emission layer properties. In order to keep the signal-to-noise ratios high, we split the sample into two parts based on a characteristic layer

parameter, calculated the corresponding mean spectra, and derived level populations from the measured line intensities for a comparison. For the split, we selected the effective height of the OH emission layer $h_{\mathrm{eff}}$, i.e. the centroid altitude weighted by the volume emission rate, as it is positively correlated with the strength of the non-LTE effects (Noll et al., 2017, 2018a). There are fewer thermalising collisions without $v'$ change at higher altitudes due to lower air densities but higher atomic oxygen mixing ratios (Noll et al., 2018b). The impact of this effect is clearly reflected by the observed higher $h_{\mathrm{eff}}$ for higher $v'$ (e.g.,

von Savigny et al., 2012), which are more affected by the hot nascent population and have lower effective lifetimes. According to the population modelling of Noll et al. (2018b) for $v' = 9$, $h_{\mathrm{eff}}$ also increases for higher $N'$.

In order to study the change of the population distribution for the different $v'$ depending on the OH emission altitude, we need adequate space-based measurements of the emission profiles to be linked with our ground-based OH level population data. This was already achieved by Noll et al. (2017) based on limb-sounding data for the Cerro Paranal region from the OH-

specific channels of the SABER radiometer (Russell et al., 1999). Here, we focus on the channel centred on 2.06 $\mu$m, which covers OH(8-6) and OH(9-7). The effective $v'$ is about 8.3 (Noll et al., 2016). Noll et al. (2017) connected the resulting $h_{\mathrm{eff}}$ for 4,496 profiles to each UVES spectrum by a weighting procedure which involved temporal differences in day of year and local time measured in a two-dimensional climatology. The approach also included the correction of differences in the solar activity, as measured by the solar radio flux (Sect. 3.2), for each UVES observation compared with the corresponding weighted $h_{\mathrm{eff}}$.

Finally, a most likely $h_{\mathrm{eff}}$ was available for each UVES spectrum. We used these data to split our sample of 553 spectra (Sect. 2) at a median $h_{\mathrm{eff}}$ of 89.2 km for the 2.06 $\mu$m OH channel. The resulting subsamples show mean $h_{\mathrm{eff}}$ of 88.7 and 89.6 km, i.e. the height difference is almost 1 km. The situation is very similar for the other OH channel at 1.64 $\mu$m representing an effective $v'$ of about 4.6 (Noll et al., 2016), where the corresponding heights are 87.3 and 88.3 km. Then, we performed the entire data





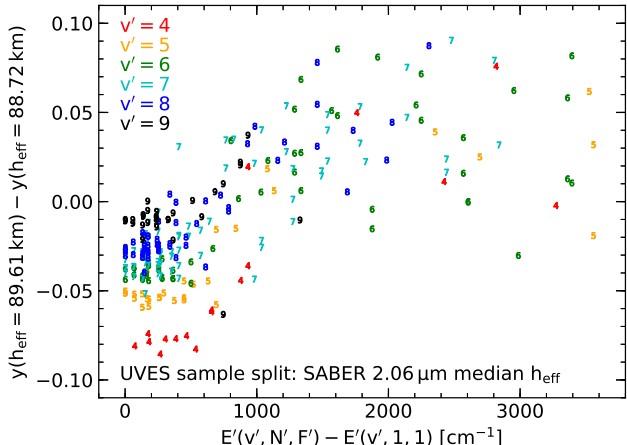

**Figure 14.** Difference in logarithmic OH level populations for UVES mean spectra based on equally sized subsamples of high and low effective emission height (89.61 and 88.72 km on average). The altitudes were derived from volume emission profiles of the SABER OH channel at $2.06\,\mu$m for the Cerro Paranal region. The data points are shown for different upper vibrational levels $v'$ (indicated by the corresponding coloured markers) as a function of the level energies in inverse centimetres relative to the lowest $v'$-related energy. The plot only shows population measurements for reliable $\Lambda$ doublets which are covered by both UVES set-ups (see Fig. 1 for the valid wavelength ranges).

analysis starting with the calculation of the mean spectra up to the derivation of the final populations. In order to minimise
systematic effects in the line measurement, the same wavelengths for the line integration and continuum derivation as for the full sample spectrum were used (see also Sect. 3.2).

    Figure 14 shows the resulting population ratios for the high and low $h_{\text{eff}}$ cases. For such a comparison, the choice of the Einstein-A coefficients does not matter. We only plot $\Delta y$ related to the 257 best-quality $\Lambda$ doublets (class 33) that are covered by all UVES spectra, i.e. reliable population ratios which are representative of the given $h_{\text{eff}}$. We can identify three energy
level regimes with respect to the population change by a rise of the OH emission layer.

    Up to about $600\,\text{cm}^{-1}$, there is a general decrease of the populations, which strongly depends on $v'$. From $v' = 4$ to 9, the decrease shrinks from about 8% ($-0.08$) to about 1%, i.e. the populations for higher $v'$ are more stable. The largest difference in $\Delta y$ is between the two lowest $v'$. The decrease of the OH intensity for a rising OH emission layer is well known (Yee et al., 1997; Melo et al., 1999; Liu and Shepherd, 2006). It is accompanied by a lower width of the layer ($-0.5\,\text{km}$ for our
subsamples) due to especially low OH production rates at the bottom side, which are caused by a depletion of ozone. As band emissions with lower $v'$ peak at lower altitudes (von Savigny et al., 2012; Noll et al., 2016), they seem to be more affected by this lack of fuel for the OH production.

    At $\Delta E'$ above $1{,}200\,\text{cm}^{-1}$, Fig. 14 shows a completely different behaviour. There is a general increase of the populations with a mean of $+0.04$ and no significant dependence on $v'$. This finding implies that the contribution of hot populations to
the total population increases with $h_{\text{eff}}$. The impact of non-LTE effects grows by a less efficient thermalisation process. The





reduced contributions from lower altitudes to the total emission certainly play an important role here since (similar to $v'$) emission related to higher $N'$ peaks higher in the atmosphere (Noll et al., 2018b). There, collisional thermalisation of the rotational level population distributions is hampered by a relatively low density of nitrogen molecules and a relatively high volume mixing ratio of $v'$-deactivating (or even OH-destroying) atomic oxygen radicals (Noll et al., 2018b). Note that the

location of the zero line in Fig. 14 is uncertain with respect to the degree of thermalisation as the increase of 4% could also be caused by a change in the OH column density. If the hot populations define the zero line as they appear to be the most stable ones (which might be supported by the lack of a $v'$ dependence), the low-$N'$ populations would further decrease.

Figure 14 is another good argument for the bimodality of rotational level population distributions. The change of $y$ for low and high $N'$ with increasing $\Delta E'$ seems to be very small. This suggests that cold and hot populations are relatively

homogeneous, which supports our two-temperature fit approach. Moreover, there is a quick transition between both populations in the relatively narrow $\Delta E'$ range between 600 and 1,200 cm$^{-1}$. In Fig. 12, this is the region where both fit components significantly contribute. Hence, a rise of the OH emission layer should there have the strongest impact on the slope of the population distribution (i.e. $T_{\mathrm{rot}}$) by a change of the relative contribution of the cold and hot populations. As $r_{\mathrm{pop},0}$ increases, there is also an effect on $\Delta T_{\mathrm{NLTE}}$ as shown in Fig. 13. Estimates of this quantity based on the populations for low and high OH

layer are relatively uncertain due to the distinctly lower number of suitable lines and the high impact of increased line intensity errors on the analysis of very small population differences. Nevertheless, we calculated $\Delta T_{\mathrm{NLTE}}$ changes of the order of a few tenths of a kelvin for the altitude difference of about 1 km. This is clearly smaller than about 1 K per kilometre, the order of magnitude from observational and modelling studies by Noll et al. (2017, 2018a, b), which might point to limitations in the study of $\Delta T_{\mathrm{NLTE}}$ variations based on two-component population fits.

## 6 Conclusions

Based on averaged high-quality high-resolution spectra from the UVES echelle spectrograph at Cerro Paranal, we performed a detailed study of OH roto-vibrational level population distributions. The mean populations for 723 $\Lambda$ doublets with upper vibrational levels $v'$ between 3 and 9 and upper rotational levels $N'$ up to 24 were investigated. In about half the cases, the doublet components were measured separately. The line wavelengths from literature (Rothman et al., 2013; Brooke et al., 2016)

turned out to be sufficiently accurate in most cases. Only a small number of lines with high $N'$ and intermediate $v'$ (especially $v' = 5$) showed deviations by more than a few picometres.

The quality of population measurements is limited by uncertainties in the Einstein-A coefficients. We investigated this issue by comparisons of populations from different transitions with the same upper state. We tested six sets of transition probabilities: Brooke et al. (2016), HITRAN (Rothman et al., 2013), van der Loo and Groenenboom (2008), Turnbull and Lowe (1989),

Langhoff et al. (1986), and Mies (1974). All sets fail in the case of Q-branch lines and the $\Lambda$-doublet components, where unexpectedly large intensity ratios are possible. The comparison of populations from P- and R-branch lines indicated relatively small errors for the coefficients of Langhoff et al. (1986), van der Loo and Groenenboom (2008), and Brooke et al. (2016), whereas those from Turnbull and Lowe (1989) are clearly the worst. The comparison of OH bands with the same $v'$ showed a



similar order of the different sets with respect to their quality. For this case, the coefficients of Brooke et al. (2016) performed

best. The widely used HITRAN data are only of intermediate and hence unsatisfying quality.

For the population analysis, we focussed on the Einstein-A coefficients from Brooke et al. (2016) due their relatively good performance and the highest number of included lines. In order to minimise the scatter in the populations, we further improved these coefficients by empirically correcting the found population discrepancies via regression lines related to $N'$ and wavelength as well as band-dependent correction factors. For the correction of the branch-related differences, we used P- and

R-branch data combined with equal weights as the reference since this strongly reduced the deviations between the different sets of Einstein-A coefficients with respect to rotational temperatures $T_{\mathrm{rot}}$, i.e. the change of the populations with increasing $N'$. The whole correction procedure lowered the discrepancies in the coefficients by more than a factor of 2 for the measured lines. Nevertheless, the development of an improved set for all lines would need a more sophisticated approach including modelling of the molecular parameters.

The resulting $v'$-dependent population distributions show clearly bimodal structures, which were convincingly reproduced by two-temperature fits only excluding steep population decreases for $v' = 8$ and 9 at the highest $N'$ with energies slightly below and above the exothermicity limit of the OH-producing hydrogen–ozone reaction, respectively. The fits show a cold population with nearly ambient temperature of about 190 K dominating at low $N'$ and a hot population with temperatures between 700 K for $v' = 9$ and 7,000 K for $v' = 4$ at high $N'$. In contrast, the ratio of the hot and cold populations at the level

with the lowest energy of a given $v'$ changes from 3 to 0.3% mainly due to a decrease of the cold component. The significant contribution of a hot population to low $N'$ causes deviations between $T_{\mathrm{rot}}$ and ambient temperature, which we estimated by fitting our two-component model for the energy levels related to the first three $P_1$-branch lines. The results indicate non-LTE contributions that increase from about 1 K for $v' = 4$ to about 5 K for $v' = 8$. The best-fit value for $v' = 9$ is even higher (about 6 K), but the fit uncertainties are by far the highest. In general, the applied approach is much more robust than the previously

used method based on comparisons of temperatures from different sources as it only weakly depends on uncertainties in the line intensities and Einstein-A coefficients. Our approach is mostly limited by the reliability of the assumption of only two Boltzmann-like population distributions. There are hints of the existence of a more complex pattern but the impact of these additional components appears to be small.

This conclusion is supported by the change of the populations due to a rise of the OH emission layer, which we studied

by the separation of the sample of spectra into two parts depending on the effective emission height as obtained from height-resolved SABER OH volume emission rates. The energy regimes up to about $600\,\mathrm{cm}^{-1}$ and above about $1,200\,\mathrm{cm}^{-1}$ relative to the lowest energy for a given $v'$ show clearly distinct variability in agreement with the energy ranges dominated by the cold and hot components in the derived population distributions. While the cold populations show a decrease, which is stronger for lower $N'$, the hot populations are relatively stable (or even increase) with increasing emission altitude. The largest measured

effect is a 12% decrease of the cold population at $v' = 4$ relative to the hot population for a height difference of almost 1 km.

The success of the two-component model for OH rotational level population distributions has implications for the thermalisation process of the highly non-thermal nascent populations. There are still high uncertainties with respect to the rate coefficients for collisions with and without change of $v'$. In particular, the modification of the rotational level population by $v'$-changing





collisions is not known. Hence, the origin of the very hot populations at high $N'$ of low $v'$ is puzzling. Consequently, there is

hope that the high-quality population data of this study can help to better understand relaxation processes in OH by detailed

modelling. This will be important knowledge with respect to the use of OH as an indicator of mesopause temperatures and for

retrievals of atomic abundances like those of oxygen.

*Data availability.* This project made use of the ESO Science Archive Facility at http://archive.eso.org. UVES Phase 3 spectra from different

observing programmes of the period from April 2000 to March 2015 were analysed. The v2.0 SABER data products used for this study were

taken from http://saber.gats-inc.com.

*Author contributions.* S. Noll has developed the project, processed the data, performed the analysis, produced the figures, and is the main

author of the paper text, where all co-authors made significant contributions. H. Winkler and O. Goussev have also influenced the design of

the study. In addition, H. Winkler has checked parts of the analysis and B. Proxauf has been involved in the post-processing of the UVES

Phase 3 products.

*Competing interests.* The authors declare that they have no conflict of interest.

*Acknowledgements.* S. Noll is financed by the project NO 1328/1-1 of the German Research Foundation (DFG). H. Winkler is funded by the

DFG project NO 404/21-1.



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
