# Peer review of "OH level populations and accuracies of Einstein-A coefficients from hundreds of measured lines"

_Atmospheric Chemistry and Physics, 2019_

## Referee Comment (RC1) · Ernesto Oliva (Referee) · 20 Dec 2019

The paper presents a very detailed analysis of OH lines intensities derived from a large set of archive spectra collected with the UVES spectrograph of the ESO-VLT telescope. It clearly shows that the molecular parameters (transition probabilities) available in the literature yield different results, none of them fully compatible with the data. The authors select the source that provides the better match and propose empirical corrections to achieve a better fit.

I am somewhat worried about the flux calibration of lines that fall in regions with significant telluric absorption, such as the (6-2)P2(12) doublet shown in Figure 9. The intrinsic widths of the airglow emission lines and of the telluric absorpion fatures could

be narrower than instrumental resolution. In such a case the correction applied is strongly dependent on model parameters such as the velocity of the airglow clouds and the assumed spectral profile of the telluric absorption. I strongly suggest to perform a sanity check, e.g. by comparing the discrepancies of Lambda-doublets (Figure 9) with the telluric absorption in the spectral region where the lines are measured.

Finally, I note that problems arising from the comparison of lines from the same upper level and with different delta-v, were also reported in Oliva et 2013 (2013A&A,..555A..78O). You may add a reference to this article.

---

## Referee Comment (RC2) · Anonymous Referee #2 · 9 Jan 2020

General Comments

This is a very well written manuscript based on a detailed analysis of one very high-quality mean spectrum derived from more than 530 individual high-quality spectra obtained during 536 hours of observations with the high-resolution Ultraviolet and Visual Echelle Spectrograph (UVES) at the telescope with a primary mirror of 8 m diameter located at Cerro Paranal in Chile. The primary target of the study is the emissions emanating from OH radicals located in a narrow layer near the Earth's mesopause. The motivation for such a study is clearly explained in the Introduction which provides a comprehensive description of the historic use of these spectra in remote sensing of the mesopause temperature and the retrieval of atomic oxygen abundances. The criteria applied in the selection of the individual spectra used to generate the very

high quality mean spectrum and the rationale behind them are well described in Section 3 as are the various corrections applied, such as the van Rhijn effect, and the solar activity correction. Historically, the calculation of the mesopause temperature was based on the assumption that the populations in the rotational energy levels are in thermal equilibrium, i.e., it is typically Trot that is determined from individual rotational transitions between different vibrational levels. The calculation of Trot and the OH populations within individual upper states depends on the Einstein-A coefficients. The wavelength range covered by the mean spectrum ($\sim$ 0.57-1.04 $\mu$m) has enabled the authors of this manuscript to examine the Einstein-A coefficients of 15 OH bands with upper vibrational levels between 3 and 9 and $\Delta v=v\hat{}'-v''$ between 3 and 6. Six of the most widely used sets of Einstein-A coefficients are tested by the very high quality mean spectrum, all of which have some failings, but having identified the optimal set (Brooke et al., 2016), the authors adopt an empirical approach to improve these coefficients. Use of the improved Einstein-A coefficients to analyse the rotational level populations indicated that the latter included a cold (largely thermalized) component together with a hot (non-thermalized) component. The values of Trot derived from OH bands, particularly those arising from higher vibrational levels, are well known to be susceptible to non-LTE effects. It is the hot populations that give rise to these non-LTE effects. These results are valuable because they quantify the extent to which different bands suffer from these non-LTE effects The authors divided the original set of UVES spectra used to produce the very high quality mean spectrum into two on the basis of the effective emission height of each individual UVES spectrum. This was already achieved in earlier work by Noll et al., (2017 and 2018a) by linking the ground-based spectra with space-based measurements of altitude profiles of the OH emissions from the SABER radiometer (Russell et al., 1999). The degree of thermalization was found to decrease with increasing altitude which could be explained by the higher fraction of the hot component. The assumptions used in the calculations, the criteria applied at each stage and the significance of the results are clearly explained. The manuscript is very well referenced and certainly deserves to be published with only very minor

textual corrections.

Specific Comments

The authors should make the empirically corrected Brooke et al. (2016) set of coefficients available as supplementary material so that others may benefit from this work.

Very minor textual corrections

This manuscript contains a lot of very detailed information: it is clear that the authors have gone to great lengths to achieve the level of precision shown in these details. I have found only one typographical error which occurs in line 630.

Line 630: refers to "553 spectra", when it should be "533 spectra" as specified in lines 90 and 95.

Lines 113-114: the final sentence of the paragraph beginning with "The smoothing . . ." is not clear.

Lines 255, 313 and 416: the authors use the word "satisfying" or "satisfyingly" when referring to the quality of Einstein-A coefficients and in line 498 when referring to the OH level populations. The words "satisfactory" or "satisfactorily" are suggested as a better alternative.

Similarly, lines 281, 302, 375, 685 include the words "unsatisfying" or "unsatisfyingly". The words "unsatisfactory" or "unsatisfactorily" would be better choices.

Line 306: suggest replace "neglection of" by "omission of" or "negligence of".

Line 498: suggest "cannot be reproduced satisfactorily" instead of "cannot satisfyingly be reproduced".

Line 662: suggest "OH emission layer there should have the strongest impact" instead of "OH emission layer should there have the strongest impact".

References

Brooke, J. S. A., Bernath, P. F., Western, C. M., Sneden, C., Afsar, M., Li, G., and Gordon, I. E.: Line strengths of rovibrational and rotational transitions in the $X2\Pi$ ground state of OH, J. Quant. Spectrosc. Radiat. Transf., 168, 142–157, https://doi.org/10.1016/j.jqsrt.2015.07.021, 745 2016 Noll, S., Kimeswenger, S., Proxauf, B., Unterguggenberger, S., Kausch, W., and Jones, A. M.: 15 years of VLT/UVES OH intensities and temperatures in comparison with TIMED/SABER data, J. Atmos. Sol.-Terr. Phys., 163, 54–69, https://doi.org/10.1016/j.jastp.2017.05.012, 2017. Noll, S., Proxauf, B., Kausch, W., and Kimeswenger, S.: Mechanisms for varying non-LTE contributions to OH rotational temperatures from measurements and modelling. I. Climatology, J. Atmos. Sol.-Terr. Phys., 175, 87–99, https://doi.org/10.1016/j.jastp.2018.05.004, 2018a. Russell, III, J. M., Mlynczak, M. G., Gordley, L. L., Tansock, J., and Esplin, R.: Overview of the SABER experiment and preliminary calibration results, in: Optical Spectroscopic Techniques and Instrumentation for Atmospheric and Space Research III, edited by Larar, A. M., vol. 3756 of SPIE Proc. Ser., pp. 277–288, https://doi.org/10.1117/12.366382, 1999.

---

## Author Comment (AC1) · 29 Jan 2020

**Response to comment by Referee #1 (Ernesto Oliva) on "OH level populations and accuracies of Einstein-A coefficients from hundreds of measured lines" by Stefan Noll et al.**

The paper presents a very detailed analysis of OH lines intensities derived from a large set of archive spectra collected with the UVES spectrograph of the ESO-VLT telescope. It clearly shows that the molecular parameters (transition probabilities) available in the literature yield different results, none of them fully compatible with the data. The authors select the source that provides the better match and propose empirical corrections to achieve a better fit.

**We thank the reviewer for preparing this positive report including two specific recommendations for improvements.**

I am somewhat worried about the flux calibration of lines that fall in regions with significant telluric absorption, such as the (6-2)P2(12) doublet shown in Figure 9. The intrinsic widths of the airglow emission lines and of the telluric absorpion fatures could be narrower than instrumental resolution. In such a case the correction applied is strongly dependent on model parameters such as the velocity of the airglow clouds and the assumed spectral profile of the telluric absorption. I strongly suggest to perform a sanity check, e.g. by comparing the discrepancies of Lambda-doublets (Figure 9) with the telluric absorption in the spectral region where the lines are measured.

**We agree that telluric absorption can be a serious issue that needs to be considered. As briefly described in the first paragraph of Sect. 3.2 and discussed in more detail in the cited papers, we performed a complex correction approach which included the derivation of the effective absorption for each line based on molecular absorption spectra with a resolving power of $10^6$ and simulated emission line profiles assuming Doppler broadening. The latter causes FWHM of about 2 pm at 800 nm, while the absorption lines are several times broader. Winds are not an issue since the speed is usually below 100 m/s, i.e. the wavelength shifts are only a few tenths of a picometre. In order to obtain realistic absorption spectra, we considered the zenith angles and the precipitable water vapour (PWV) during each observation. The PWV was derived from the water vapour absorption in the corresponding astronomical target spectrum, which usually showed a sufficiently bright continuum.**

**The corrected line intensities are usually quite reliable. Only in the wings of very strong absorption lines where the transmission gradients are high, there might be significant uncertainties. However, these cases are rare in the wavelength range of UVES, where weak absorption dominates. For the measured lines, the mean transmission is 97%. Only 6% of the Λ doublets were absorbed by more than 10%. Hence, after the correction, which can reduce the absorption by up to an order of magnitude, the impact of telluric absorption on most lines is negligible. We add this information to the end of the first paragraph of Sect. 3.2:**
**"… spectra were averaged. The resulting mean absorption of the measured Λ doublets was 3% and only 6% of the doublets were attenuated by more than 10%. Hence, the related intensity uncertainty after the correction, which can reduce the absorption by up to an order of magnitude, is negligible for most lines."**

**The distribution of data points in Fig. 9 depending on the transmission differences between both Λ doublet components can be used for a check. 8 of the 140 P-branch-related doublets show a transmission difference of more than 10%. However, their distribution well agrees with the one for the full sample. There is no increased scatter, which is confirmed by a regression analysis. Also note that the possible impact of absorption lines close to the considered emission line (partial blends) on the quality of the intensity measurement has been considered via the quality class. While 6% of all measured doublets show absorption of more than 10%, this percentage is only 3% for the best class 33.**

Finally, I note that problems arising from the comparison of lines from the same upper level and with different delta-v, were also reported in Oliva et 2013 (2013A&A,..555A..78O). You may add a reference to this article.

**We thank the reviewer for mentioning this paper with important results based on OH population comparisons. We have missed to consider it for the submitted manuscript, but we will discuss it in the revised version.**

**The corresponding text is entered at L. 381:**
**"… transition probabilities. Population comparisons for OH lines from near-infrared bands with low Δv mostly not covered by UVES were performed by Oliva et al. (2013) based on observations between 0.95 and 2.4 µm with the high-resolution echelle spectrograph GIANO at the Telescopio Nazionale Galileo at the La Palma Observatory in Spain. The results show clear discrepancies between populations derived from lines of bands with Δv = 2, 3 and 4 for the Einstein-A coefficients from van der Loo and Groenenboom (2007). Interestingly, the corresponding trend with wavelength displayed in Fig. 8 seems to be reversed for bands at longer wavelengths. In general, ..."**

**As a consequence, some text later in the paper will be redundant:**
**"which are not accessible by UVES" in L. 383 can be removed and the GIANO description in L. 528-530 can be shortened:**
**"Two-component fits were previously performed by Oliva et al. (2015) based on a near-infrared GIANO spectrum with a resolving power of 32,000 taken during 2 hours with the spectrograph directly pointing to the night sky at the La Palma Observatory. The investigated lines ..."**

---

## Author Comment (AC2) · 29 Jan 2020

**Response to comment by Anonymous Referee #2 on "OH level populations and accuracies of Einstein-A coefficients from hundreds of measured lines" by Stefan Noll et al.**

General Comments

This is a very well written manuscript based on a detailed analysis of … The assumptions used in the calculations, the criteria applied at each stage and the significance of the results are clearly explained. The manuscript is very well referenced and certainly deserves to be published with only very minor textual corrections.

**We thank the reviewer for the careful reading of the manuscript, which is illustrated by the detailed summary of the paper content in the review (not reproduced here because of its length). In the view of the reviewer's efforts, we are pleased to see the very positive evaluation of the paper.**

Specific Comments

The authors should make the empirically corrected Brooke et al. (2016) set of coefficients available as supplementary material so that others may benefit from this work.

**We agree that the modified Einstein-A coefficients should be published. On the website of the discussion paper, they can already be downloaded included in a supplement. The zip archive contains all data that are needed to reproduce the figures related to the OH line intensities and their analysis.**

Very minor textual corrections

This manuscript contains a lot of very detailed information: it is clear that the authors have gone to great lengths to achieve the level of precision shown in these details. I have found only one typographical error which occurs in line 630.

**We thank the reviewer for spotting the wrong number and several linguistic issues. We have corrected everything as proposed.**

Line 630: refers to "553 spectra", when it should be "533 spectra" as specified in lines 90 and 95.

**Done.**

Lines 113-114: the final sentence of the paragraph beginning with "The smoothing ..." is not clear.

**In the revised version, we will write "The rounded edges of the sample-related steps in the histogram reflect the variation in the wavelength positioning of a certain set-up.". We hope that "rounded edges" is clearer than "smoothing".**

Lines 255, 313 and 416: the authors use the word "satisfying" or "satisfyingly" when referring to the quality of Einstein-A coefficients and in line 498 when referring to the OH level populations. The words "satisfactory" or "satisfactorily" are suggested as a better alternative.

**Done.**

Similarly, lines 281, 302, 375, 685 include the words "unsatisfying" or "unsatisfyingly". The words "unsatisfactory" or "unsatisfactorily" would be better choices.

**Done.**

Line 306: suggest replace "neglection of" by "omission of" or "negligence of".

**Done.**

Line 498: suggest "cannot be reproduced satisfactorily" instead of "cannot satisfyingly be reproduced".

**Done.**

Line 662: suggest "OH emission layer there should have the strongest impact" instead of "OH emission layer should there have the strongest impact".

**Done.**